# WHEN SOURCE-FREE DOMAIN ADAPTATION MEETS LEARNING WITH NOISY LABELS

**Li Yi**[1,*]   **Gezheng Xu**[2,*]   **Pengcheng Xu**[2]   **Jiaqi Li**[2]   **Ruizhi Pu**[2]
**Charles Ling**[2]   **A. Ian McLeod**[1]   **Boyu Wang**[1,2,†]
[1]Department of Statistical and Actuarial Sciences   [2]Department of Computer Science
University of Western Ontario
`{lyi7,gxu86,pxu67,jli3779,rpu2,charles.ling,aimcleod}@uwo.ca`
` bwang@csd.uwo.ca`

## ABSTRACT

Recent state-of-the-art source-free domain adaptation (SFDA) methods have focused on learning meaningful cluster structures in the feature space, which have succeeded in adapting the knowledge from source domain to unlabeled target domain without accessing the private source data. However, existing methods rely on the pseudo-labels generated by source models that can be noisy due to domain shift. In this paper, we study SFDA from the perspective of learning with label noise (LLN). Unlike the label noise in the conventional LLN scenario, we prove that the label noise in SFDA follows a different distribution assumption. We also prove that such a difference makes existing LLN methods that rely on their distribution assumptions unable to address the label noise in SFDA. Empirical evidence suggests that only marginal improvements are achieved when applying the existing LLN methods to solve the SFDA problem. On the other hand, although there exists a fundamental difference between the label noise in the two scenarios, we demonstrate theoretically that the early-time training phenomenon (ETP), which has been previously observed in conventional label noise settings, can also be observed in the SFDA problem. Extensive experiments demonstrate significant improvements to existing SFDA algorithms by leveraging ETP to address the label noise in SFDA.

## 1  INTRODUCTION

Deep learning demonstrates strong performance on various tasks across different fields. However, it is limited by the requirement of large-scale labeled and independent, and identically distributed (i.i.d.) data. Unsupervised domain adaptation (UDA) is thus proposed to mitigate the distribution shift between the labeled source and unlabeled target domain. In view of the importance of data privacy, it is crucial to be able to adapt a pre-trained source model to the unlabeled target domain without accessing the private source data, which is known as Source Free Domain Adaptation (SFDA).

The current state-of-the-art SFDA methods (Liang et al., 2020; Yang et al., 2021a;b) mainly focus on learning meaningful cluster structures in the feature space, and the quality of the learned cluster structures hinges on the reliability of pseudo labels generated by the source model. Among these methods, SHOT (Liang et al., 2020) purifies pseudo labels of target data based on nearest centroids, and then the purified pseudo labels are used to guide the self-training. G-SFDA (Yang et al., 2021b) and NRC (Yang et al., 2021a) further refine pseudo labels by encouraging similar predictions to the data point and its neighbors. For a single target data point, when most of its neighbors are correctly predicted, these methods can provide an accurate pseudo label to the data point. However, as we illustrate the problem in Figure 1i(a-b), when the majority of its neighbors are incorrectly predicted to a category, it will be assigned with an incorrect pseudo label, misleading the learning of cluster structures. The experimental result on VisDA (Peng et al., 2017), shown in Figure 1ii, further verifies this phenomenon. By directly applying the pre-trained source model on each target domain instance

---

*Equal contribution
†Corresponding author

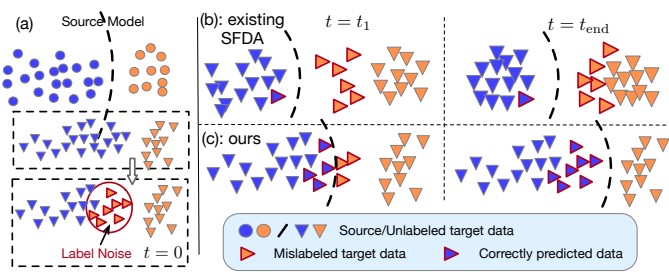 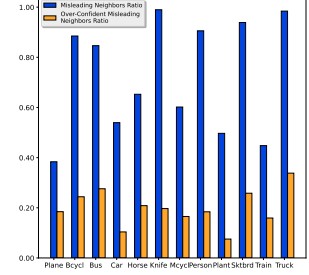

(i) Overview of the SFDA problem and our method  (ii) Neighbors Label Noise Analysis On VisDA

Figure 1: (i) (a) The SFDA problem can be formulated as an LLN problem. (b) The existing SFDA algorithms using the local cluster information cannot address label noise due to the unbounded label noise (Section 3). (c) We prove that ETP exists in SFDA, which can be leveraged to address the unbounded label noise (Section 4). (ii) Observed Label Noise Phenomena on VisDA dataset.

(central instance), we collect its neighbors and evaluate their quality. We observed that for each class a large proportion of the neighbors are *misleading* (i.e., the neighbors' pseudo labels are different from the central instance's true label), some even with high confidence (e.g., the *over-confident misleading neighbors* whose prediction score is larger than 0.75). Based on this observation, we can conclude that: (1) the pseudo labels leveraged in current SFDA methods can be heavily noisy; (2) some pseudo-label purification methods utilized in SFDA, which severely rely on the quality of the pseudo label itself, will be affected by such label noise, and the prediction error will accumulate as the training progresses. More details can be found in Appendix A.

In this paper, we address the aforementioned problem by formulating SFDA as *learning with label noise* (LLN). Unlike existing studies that heuristically rely on cluster structures or neighbors, we investigate the properties of label noise in SFDA and show that there is an intrinsic discrepancy between the SFDA and the LLN problems. Specifically, in conventional LLN scenarios, the label noise is generated by human annotators or image search engines (Patrini et al., 2017; Xiao et al., 2015; Xia et al., 2020a), where the underlying distribution assumption is that the mislabeling rate for a sample is bounded. However, in the SFDA scenarios, the label noise is generated by the source model due to the distribution shift, where we prove that the mislabeling rate for a sample is much higher, and can approach 1. We term the former label noise in LLN as *bounded label noise* and the latter label noise in SFDA as *unbounded label noise*. Moreover, we theoretically show that most existing LLN methods, which rely on bounded label noise assumption, are unable to address the label noise in SFDA due to the fundamental difference (Section 3).

To this end, we leverage *early-time training phenomenon* (ETP) in LLN to address the unbounded label noise and to improve the efficiency of existing SFDA algorithms. Specifically, ETP indicates that classifiers can predict mislabeled samples with relatively high accuracy during the early learning phase before they start to memorize the mislabeled data (Liu et al., 2020). Although ETP has been previously observed in, it has only been studied in the bounded random label noise in the conventional LLN scenarios. In this work, we theoretically and empirically show that ETP still exists in the unbounded label noise scenario of SFDA. Moreover, we also empirically justify that existing SFDA algorithms can be substantially improved by leveraging ETP, which opens up a new avenue for SFDA. As an instantiation, we incorporate a simple early learning regularization (ELR) term (Liu et al., 2020) with existing SFDA objective functions, achieving consistent improvements on four different SFDA benchmark datasets. As a comparison, we also apply other existing LLN methods, including Generalized Cross Entropy (GCE) (Zhang & Sabuncu, 2018), Symmetric Cross Entropy Learning (SL) (Wang et al., 2019b), Generalized Jensen-Shannon Divergence (GJS) (Englesson & Azizpour, 2021) and Progressive Label Correction (PLC) (Zhang et al., 2021), to SFDA. Our empirical evidence shows that they are inappropriate for addressing the label noise in SFDA. This is also consistent with our theoretical results (Section 4).

Our main contribution can be summarized as: (1) We establish the connection between the SFDA and the LLN. Compared with the conventional LLN problem that assumes bounded label noise, the problem in SFDA can be viewed as the problem of LLN with the unbounded label noise. (2)

We theoretically and empirically justify that ETP exists in the unbounded label noise scenario. On the algorithmic side, we instantiate our analysis by simply adding a regularization term into the SFDA objective functions. (3) We conduct extensive experiments to show that ETP can be utilized to improve many existing SFDA algorithms by a large margin across multiple SFDA benchmarks.

## 2 RELATED WORK

**Source-free domain adaptation.** Recently, SFDA are studied for data privacy. The first branch of research is to leverage the target pseudo labels to conduct self-training to implicitly achieve adaptation (Liang et al., 2021; Tanwisuth et al., 2021; Ahmed et al., 2021; Yang et al., 2021b). SHOT (Liang et al., 2020) introduces k-means clustering and mutual information maximization strategy for self-training. NRC (Yang et al., 2021a) further investigates the neighbors of target clusters to improve the accuracy of pseudo labels. These studies more or less involve pseudo-label purification processes, but they are primarily heuristic algorithms and suffer from the previously mentioned label noise accumulation problem. The other branch is to utilize the generative model to synthesize target-style training data (Qiu et al., 2021; Liu et al., 2021b). Some methods also explore the SFDA algorithms in various settings. USFDA (Kundu et al., 2020a) and FS (Kundu et al., 2020b) design methods for universal and open-set UDA. In this paper, we regard SFDA as the LLN problem. We aim to explore what category of noisy labels exists in SFDA and to ameliorate such label noise to improve the performance of current SFDA algorithms.

**Learning with label noise.** Existing methods for training neural networks with label noise focus on symmetric, asymmetric, and instance-dependent label noise. For example, a branch of research focuses on leveraging noise-robust loss functions to cope with the symmetric and asymmetric noise, including GCE (Zhang & Sabuncu, 2018), SL (Wang et al., 2019b), NCE (Ma et al., 2020), and GJS (Englesson & Azizpour, 2021), which have been proven effective in bounded label noise. On the other hand, CORES (Cheng et al., 2020) and CAL (Zhu et al., 2021) are shown useful in mitigating instance-dependent label noise. These methods are only tailed to conventional LLN settings. Recently, Liu et al. (2020) has studied early-time training phenomenon (ETP) in conventional label noise scenarios and proposes a regularization term ELR to exploit the benefits of ETP. PCL (Zhang et al., 2021) is another conventional LLN algorithm utilizing ETP, but it cannot maintain the exploit of ETP in SFDA as memorizing noisy labels is much faster in SFDA. Our contributions are: (1) We theoretically and empirically study ETP in the SFDA scenario. (2) Based on an in depth analysis of many existing LLN methods (Zhang & Sabuncu, 2018; Wang et al., 2019b; Englesson & Azizpour, 2021; Zhang et al., 2021), we demonstrate that ELR is useful for many SFDA problems.

## 3 LABEL NOISE IN SFDA

The presence of label noise on training datasets has been shown to degrade the model performance (Malach & Shalev-Shwartz, 2017; Han et al., 2018). In SFDA, existing algorithms rely on pseudo-labels produced by the source model, which are inevitably noisy due to the domain shift. The SFDA methods such as Liang et al. (2020); Yang et al. (2021a;b) cannot tackle the situation when some target samples and their neighbors are all incorrectly predicted by the source model. In this section, we formulate the SFDA as the problem of LLN to address this issue. We assume that the source domain $\mathcal{D}_S$ and the target domain $\mathcal{D}_T$ follow two different underlying distributions over $\mathcal{X} \times \mathcal{Y}$, where $\mathcal{X}$ and $\mathcal{Y}$ are respectively the input and label spaces. In the SFDA setting, we aim to learn a target classifier $f(\mathbf{x}; \theta) : \mathcal{X} \to \mathcal{Y}$ only with a pre-trained model $f_S(\mathbf{x})$ on $\mathcal{D}_S$ and a set of unlabeled target domain observations drawn from $\mathcal{D}_T$. We regard the incorrectly assigned pseudo-labels as noisy labels. Unlike the "bounded label noise" assumption in the conventional LLN domain, we will show that the label noise in SFDA is *unbounded*. We further prove that most existing LLN methods that rely on the bounded assumption cannot address the label noise in SFDA due to the difference.

**Label noise in conventional LLN settings:** In conventional label noise settings, the injected noisy labels are collected by either human annotators or image search engines (Lee et al., 2018; Li et al., 2017; Xiao et al., 2015). The label noise is usually assumed to be either independent of instances (i.e., symmetric label noise or asymmetric label noise) (Patrini et al., 2017; Liu & Tao, 2015; Xu et al., 2019b) or dependent of instances (i.e., instance-dependent label noise) (Berthon et al., 2021; Xia et al., 2020b). The underling assumption for them is that a sample $\mathbf{x}$ has the highest probability of being in the correct class $y$, i.e., $\Pr[\tilde{Y} = i|Y = i, X = x] > \Pr[\tilde{Y} = j|Y = i, X = x], \forall x \in \mathcal{X}, i \neq j,$

where $\tilde{Y}$ is the noisy label and $Y$ is the ground-truth label for input $X$. Equivalently, it assumes a bounded noise rate. For example, given an image to annotate, the mislabeling rate for the image is bounded by a small number, which is realistic in conventional LLN settings (Xia et al., 2020b; Cheng et al., 2020). When the label noise is generated by the source model, the underlying assumption of these types of label noise does not hold.

**Label noise in SFDA:** As for the label noise generated by the source model, mislabeling rate for an image can approach 1, that is, $\Pr[\tilde{Y} = j | Y = i, X = x] \rightarrow 1$, $\exists \mathcal{S} \subset \mathcal{X}$, $\forall x \in \mathcal{S}, i \neq j$. To understand that the label noise in SFDA is unbounded, we consider a two-component Multivariate Gaussian mixture distribution with equal priors for both domains. Let the first component ($y = 1$) of the source domain distribution $\mathcal{D}_S$ be $\mathcal{N}(\boldsymbol{\mu_1}, \sigma^2 \mathbf{I}_d)$, and the second component ($y = -1$) of $\mathcal{D}_S$ be $\mathcal{N}(\boldsymbol{\mu_2}, \sigma^2 \mathbf{I}_d)$, where $\boldsymbol{\mu_1}, \boldsymbol{\mu_2} \in \mathbb{R}^d$ and $\mathbf{I}_d \in \mathbb{R}^{d \times d}$. For the target domain distribution $\mathcal{D}_T$, let the first component ($y = 1$) of $\mathcal{D}_T$ be $\mathcal{N}(\boldsymbol{\mu_1} + \boldsymbol{\Delta}, \sigma^2 \mathbf{I}_d)$, and the second component ($y = -1$) of $\mathcal{D}_T$ be $\mathcal{N}(\boldsymbol{\mu_2} + \boldsymbol{\Delta}, \sigma^2 \mathbf{I}_d)$, where $\boldsymbol{\Delta} \in \mathbb{R}^d$ is the shift of the two domains. Notice that the domain shift considered is a general shift and it has been studied in Stojanov et al. (2021); Zhao et al. (2019), where we also illustrate the domain shift in Figure 9 in supplementary material.

Let $f_S$ be the optimal source classifier. First, we build the relationship between the mislabeling rate for target data and the domain shift:

$$\Pr_{(\mathbf{x}, y) \sim \mathcal{D}_T}[f_S(\mathbf{x}) \neq y] = \frac{1}{2} \Phi(-\frac{d_1}{\sigma}) + \frac{1}{2} \Phi(-\frac{d_2}{\sigma}), \tag{1}$$

where $d_1 = \left\| \frac{\boldsymbol{\mu_2} - \boldsymbol{\mu_1}}{2} - \mathbf{c} \right\| \operatorname{sign}(\left\| \frac{\boldsymbol{\mu_2} - \boldsymbol{\mu_1}}{2} \right\| - \|\mathbf{c}\|)$, $d_2 = \left\| \frac{\boldsymbol{\mu_2} - \boldsymbol{\mu_1}}{2} + \mathbf{c} \right\|$, $\mathbf{c} = \alpha(\boldsymbol{\mu_2} - \boldsymbol{\mu_1})$, $\alpha = \frac{\boldsymbol{\Delta}^\top (\boldsymbol{\mu_2} - \boldsymbol{\mu_1})}{\|\boldsymbol{\mu_2} - \boldsymbol{\mu_1}\|^2}$ is the magnitude of domain shift, and $\Phi$ is the standard normal cumulative distribution function. Eq. (1) shows that the magnitude of the domain shift inherently controls the mislabeling error for target data. This mislabeling rate increases as the magnitude of the domain shift increases. We defer the proof and details to Appendix B.

More importantly, we characterize that the label noise is unbounded among these mislabeled samples.

**Theorem 3.1.** *Without loss of generality, we assume that the* $\boldsymbol{\Delta}$ *is positively correlated with the vector* $\boldsymbol{\mu_2} - \boldsymbol{\mu_1}$, *i.e.,* $\boldsymbol{\Delta}^\top (\boldsymbol{\mu_2} - \boldsymbol{\mu_1}) > 0$. *For* $(\mathbf{x}, y) \sim \mathcal{D}_T$, *if* $\mathbf{x} \in \mathbf{R}$, *then*

$$\Pr[f_S(\mathbf{x}) \neq y] \geq 1 - \delta, \tag{2}$$

*where* $\delta \in (0, 1)$ *(i.e.,* $\delta = 0.01$*),* $\mathbf{R} = \mathbf{R_1} \bigcap \mathbf{R_2}$, $\mathbf{R_1} = \{\mathbf{x} : \|\mathbf{x} - \boldsymbol{\mu_1} - \boldsymbol{\Delta}\| \leq \sigma(\frac{\sqrt{d}}{2} - \frac{\log \frac{1-\delta}{\delta}}{\sqrt{d}})\}$, *and* $\mathbf{R_2} = \{\mathbf{x} : \mathbf{x}^\top \mathbf{1}_d > (\sigma d + 2\boldsymbol{\mu_1}^\top \mathbf{1}_d)/2\}$. *Meanwhile,* $\mathbf{R}$ *is non-empty when* $\alpha > (\log \frac{1-\delta}{\delta})/d$, *where* $\alpha = \frac{\boldsymbol{\Delta}^\top (\boldsymbol{\mu_2} - \boldsymbol{\mu_1})}{\|\boldsymbol{\mu_2} - \boldsymbol{\mu_1}\|^2} > 0$ *is the magnitude of the domain shift along the direction* $\boldsymbol{\mu_2} - \boldsymbol{\mu_1}$.

Conventional LLN methods assume that the label noise is bounded: $\Pr[f_H(\mathbf{x}) \neq y] < m$, $\forall (\mathbf{x}, y) \sim \mathcal{D}_T$, where $f_H$ is the labeling function, and $m = 0.5$ if the number of clean samples of each component are the same (Cheng et al., 2020). However, Theorem 3.1 indicates that the label noise generated by the source model is unbounded for any $\mathbf{x} \in \mathbf{R}$. In practice, region $\mathbf{R}$ is non-empty as neural networks are usually trained on high dimensional data such that $d \gg 1$, so $\alpha > (\log \frac{1-\delta}{\delta})/d \rightarrow 0$ is easy to satisfy. The probability measure on $\mathbf{R} = \mathbf{R_1} \bigcap \mathbf{R_2}$ (i.e., $\Pr_{(\mathbf{x}, y) \sim \mathcal{D}_T}[\mathbf{x} \in \mathbf{R}]$) increases as the magnitude of the domain shift $\alpha$ increases, meaning more data points contradict the conventional LLN assumption. More details can be found in Appendix C.

Given that the unbounded label noise exists in SFDA, the following Lemma establishes that many existing LLN methods (Wang et al., 2019b; Ghosh et al., 2017; Englesson & Azizpour, 2021; Ma et al., 2020), which rely on the bounded assumption, are *not* noise tolerant in SFDA.

**Lemma 3.2.** *Let the risk of the function* $h : \mathcal{X} \rightarrow \mathcal{Y}$ *under the clean data be* $R(h) = \mathbb{E}_{\mathbf{x}, y}[\ell_{LLN}(h(\mathbf{x}), y)]$, *and the risk of* $h$ *under the noisy data be* $\widetilde{R}(h) = \mathbb{E}_{\mathbf{x}, \tilde{y}}[\ell_{LLN}(h(\mathbf{x}), \tilde{y})]$, *where the noisy data follows the unbounded assumption, i.e.,* $\Pr[\tilde{y} \neq y | \mathbf{x} \in \mathbf{R}] = 1 - \delta$ *for a subset* $\mathbf{R} \subset \mathcal{X}$ *and* $\delta \in (0, 1)$. *Then the global minimizer* $\tilde{h}^\star$ *of* $\widetilde{R}(h)$ *disagrees with the global minimizer* $h^\star$ *of* $R(h)$ *on data points* $\mathbf{x} \in \mathbf{R}$ *with a high probability at least* $1 - \delta$.

We denote $\ell_{\text{LLN}}$ by the existing noise-robust loss based LLN methods in Wang et al. (2019b); Ghosh et al. (2017); Englesson & Azizpour (2021); Ma et al. (2020). When the noisy data follows the bounded assumption, these methods are noise tolerant as the minimizer $\tilde{h}^\star$ converges to the minimizer $h^\star$ with a high probability. We defer the details and proof of the related LLN methods to Appendix D.

## 4   LEARNING WITH LABEL NOISE IN SFDA

Given a fundamental difference between the label noise in SFDA and the label noise in conventional LLN scenarios, existing LLN methods, whose underlying assumption is bounded label noise, cannot be applied to solve the label noise in SFDA. This section focuses on investigating how to address the unbounded label noise in SFDA.

Motivated by the recent studies Liu et al. (2020); Arpit et al. (2017), which observed an early-time training phenomenon (ETP) on noisy datasets with bounded random label noise, we find that ETP does not rely on the bounded random label noise assumption, and it can be generalized to the unbounded label noise in SFDA. ETP describes the training dynamics of the classifier that preferentially fits the clean samples and therefore has higher prediction accuracy for mislabeled samples during the early-training stage. Such training characteristics can be very beneficial for SFDA problems in which we only have access to the source model and the highly noisy target data. To theoretically prove ETP in the presence of unbounded label noise, we first describe the problem setup.

We still consider a two-component Gaussian mixture distribution with equal priors. We denote $y$ by the true label for $\mathbf{x}$, and assume it is a balanced sample from $\{-1, +1\}$. The instance $\mathbf{x}$ is sampled from the distribution $\mathcal{N}(y\boldsymbol{\mu}, \ \sigma\mathbf{1}_d)$, where $\|\boldsymbol{\mu}\| = 1$. We denote $\tilde{y}$ by the noisy label for $\mathbf{x}$. We observe that the label noise generated by the source model is close to the decision boundary revealed in Theorem 3.1. So, to assign the noisy labels, we let $\tilde{y} = y\beta(\mathbf{x}, y)$, where $\beta(\mathbf{x}, y) = \text{sign}(\mathbb{1}\{y\mathbf{x}^\top\boldsymbol{\mu} > r\} - 0.5)$ is the label flipping function, and $r$ controls the mislabeling rate. If $\beta(\mathbf{x}, y) < 1$, then the data point $\mathbf{x}$ is mislabeled. Meanwhile, the label noise is unbounded by adopting the label flipping function $\beta(\mathbf{x}, y)$: $\Pr[\tilde{y} \neq y|y\mathbf{x}^\top\boldsymbol{\mu} \leq r] = 1$, where $\mathbf{R} = \{\mathbf{x} : y\mathbf{x}^\top\boldsymbol{\mu} \leq r\}$.

We study the early-time training dynamics of gradient descent on the linear classifier. The parameter $\theta$ is learned over the unbounded label noise data $\{x_i, \tilde{y}_i\}_{i=1}^n$ with the following logistic loss function:

$$\mathcal{L}(\theta_{t+1}) = \frac{1}{n}\sum_{i=1}^n \log\left(1 + \exp\left(-\tilde{y}_i\theta_{t+1}^\top\mathbf{x}_i\right)\right),$$

where $\theta_{t+1} = \theta_t - \eta\nabla_\theta\mathcal{L}(\theta_t)$, and $\eta$ is the learning rate. Then the following theorem builds the connection between the prediction accuracy for mislabeled samples at an early-training time $T$.

**Theorem 4.1.** *Let $B = \{\mathbf{x} : \tilde{y} \neq y\}$ be a set of mislabeled samples. Let $\kappa(B; \theta)$ be the prediction accuracy calculated by the ground-truth labels and the predicted labels by the classifier with parameter $\theta$ for mislabeled samples. If at most half of the samples are mislabeled ($r < 1$), then there exists a proper time $T$ and a constant $c_0 > 0$ such that for any $0 < \sigma < c_0$ and $n \to \infty$, with probability $1 - o_p(1)$:*

$$\kappa(B; \theta_T) \geq 1 - \exp\{-\frac{1}{200}g(\sigma)^2\}, \tag{3}$$

*where $g(\sigma) = \frac{\text{Erf}[\frac{1-r}{\sqrt{2}\sigma}]}{2(1+2\sigma)\sigma} + \frac{\exp\left(-\frac{(r-1)^2}{2\sigma^2}\right)}{\sqrt{2\pi}(1+2\sigma)} > 0$ is a monotone decreasing function that $g(\sigma) \to \infty$ as $\sigma \to 0$, and $\text{Erf}[x] = \frac{2}{\sqrt{\pi}}\int_0^x e^{-t^2}\,dt$.*

The proof is provided in Appendix E. Compared to ETP found in Liu et al. (2020), where the label noise is assumed to be bounded, Theorem 4.1 presents that ETP also exists even though the label noise is unbounded. At a proper time T, the classifier trained by the gradient descent algorithm can provide accurate predictions for mislabeled samples, where its accuracy is lower bounded by a function of the variance of clusters $\sigma$. When $\sigma \to 0$, the predictions of all mislabeled samples equal to their ground-truth labels (i.e., $\kappa(B; \theta_T) \to 1$). When the classifier is trained for a sufficiently long time, it will gradually memorize mislabeled data. The predictions of mislabeled samples are equivalent to their incorrect labels instead of their ground-truth labels (Liu et al., 2020; Maennel et al., 2020). Based on these insights, the memorization of mislabeled data can be alleviated by leveraging their predicted labels during the early-training time.

To leverage the predictions during the early-training time, we adopt a recently established method, early learning regularization (ELR) (Liu et al., 2020), which encourages model predictions to stick to the early-time predictions for $\mathbf{x}$. Since ETP exists in the scenarios of the unbounded label noise, ELR can be applied to solve the label noise in SFDA. The regularization is given by:

$$\mathcal{L}_{\text{ELR}}(\theta_t) = \log(1 - \bar{y}_t^\top f(\mathbf{x}; \theta_t)), \tag{4}$$

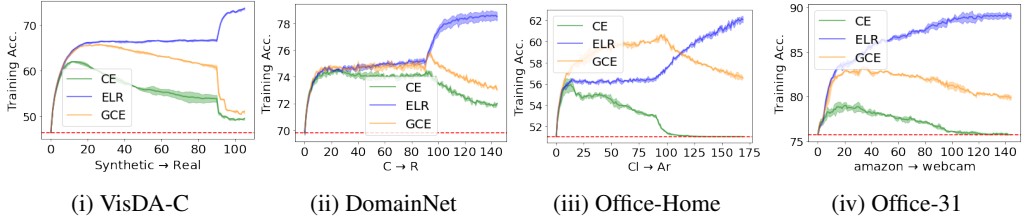

Figure 2: Training accuracy on various target domains. The source models initialize the classifiers and annotate unlabeled target data. As the classifiers memorize the unbounded label noise very fast, for the first 90 steps, we evaluate the prediction accuracy on target data every batch, and one step represents one training batch. After the 90 steps, we evaluate the prediction accuracy for every 0.3 epoch, shown as one step. We use the CE, GCE, and ELR to train the classifiers on the labeled target data, shown in solid green lines, solid orange lines, and solid blue lines, respectively. The dotted red line represents the accuracy of labeling target data. Eventually, the classifiers memorize the label noise, and the prediction accuracy equals the labeling accuracy (shown in (iii-iv)). Additional results on transfer pairs can be found in Appendix F.

where we overload $f(\mathbf{x}; \theta_t)$ to be the probabilistic output for the sample $\mathbf{x}$, and $\bar{y}_t = \beta \bar{y}_{t-1} + (1 - \beta) f(\mathbf{x}; \theta_t)$ is the moving average prediction for $\mathbf{x}$, where $\beta$ is a hyperparameter. To see how ELR prevents the model from memorizing the label noise, we calculate the gradient of Eq. (4) with respect to $f(\mathbf{x}; \theta_t)$, which is given by:

$$\frac{\mathrm{d}\mathcal{L}_{\mathrm{ELR}}(\theta_t)}{\mathrm{d}f(\mathbf{x}; \theta_t)} = -\frac{\bar{y}_t}{1 - \bar{y}_t^\top f(\mathbf{x}; \theta_t)}.$$

Note that minimizing Eq. (4) forces $f(\mathbf{x}; \theta_t)$ to close to $\bar{y}_t$. When $\bar{y}_t$ is aligned better with $f(\mathbf{x}; \theta_t)$, the magnitude of the gradient becomes larger. It makes the gradient of aligning $f(\mathbf{x}; \theta_t)$ with $\bar{y}_t$ overwhelm the gradient of other loss terms that align $f(\mathbf{x}; \theta_t)$ with noisy labels. As the training progresses, the moving averaged predictions $\bar{y}_t$ for target samples gradually approach their ground-truth labels till the time $T$. Therefore, Eq. (4) prevents the model from memorizing the label noise by forcing the model predictions to stay close to these moving averaged predictions $\bar{y}_t$, which are very likely to be ground-truth labels.

Some existing LLN methods propose to assign pseudo labels to data or require two-stage training for label noise (Cheng et al., 2020; Zhu et al., 2021; Zhang et al., 2021). Unlike these LLN methods, Eq. (4) can be easily embedded into any existing SFDA algorithms without conflict. The overall objective function is given by:

$$\mathcal{L} = \mathcal{L}_{\mathrm{SFDA}} + \lambda \mathcal{L}_{\mathrm{ELR}}, \tag{5}$$

where $\mathcal{L}_{\mathrm{SFDA}}$ is *any* SFDA objective function, and $\lambda$ is a hyperparameter.

**Empirical Observations on Real-World Datasets.** We empirically verify that target classifiers have higher prediction accuracy for target data during the early training and adaptation stage. We propose leveraging this benefit to prevent the classifier from memorizing the noisy labels. The observations are shown in Figure 2. The parameters of classifiers are initialized by source models. Labels of target data are annotated by the initialized classifiers. We train the target classifiers on target data with the standard cross-entropy (CE) loss and the generalized cross-entropy (GCE) loss, a well-known noise-robust loss widely leveraged in bounded LLN scenarios. The solid green, orange and blue lines represent the training accuracy of optimizing the classifiers with CE loss, GCE loss, and ELR loss, respectively. The dotted red lines represent the labeling accuracy of the initialized classifiers. Considering that the classifiers memorize the unbounded label noise very fast, we evaluate the prediction accuracy on target data every batch for the first 90 steps. After 90 steps, we evaluate the prediction accuracy for every 0.33 epoch. The green lines show that ETP exists in SFDA, which is consistent with our theoretical result. Meanwhile, in all scenarios, green and orange lines show that classifiers provide higher prediction accuracy during the first a few iterations. After a few iterations, they start to memorize the label noise even with noise-robust loss (e.g., GCE). Eventually, the classifiers are expected to memorize the whole datasets. For conventional LLN settings, it has been empirically verified that it takes a *much longer* time before classifiers start memorizing the label noise (Liu et al., 2020; Xia et al., 2020a). We provide further analysis in Appendix H. We highlight

Table 1: Accuracies (%) on Office-Home for ResNet50-based methods.

| Method | SF | Ar→Cl | Ar→Pr | Ar→Rw | Cl→Ar | Cl→Pr | Cl→Rw | Pr→Ar | Pr→Cl | Pr→Rw | Rw→Ar | Rw→Cl | Rw→Pr | Avg |
|---|---|---|---|---|---|---|---|---|---|---|---|---|---|---|
| MCD (Saito et al., 2018b) | ✗ | 48.9 | 68.3 | 74.6 | 61.3 | 67.6 | 68.8 | 57.0 | 47.1 | 75.1 | 69.1 | 52.2 | 79.6 | 64.1 |
| CDAN (Long et al., 2018) | ✗ | 50.7 | 70.6 | 76.0 | 57.6 | 70.0 | 70.0 | 57.4 | 50.9 | 77.3 | 70.9 | 56.7 | 81.6 | 65.8 |
| SAFN (Xu et al., 2019a) | ✗ | 52.0 | 71.7 | 76.3 | 64.2 | 69.9 | 71.9 | 63.7 | 51.4 | 77.1 | 70.9 | 57.1 | 81.5 | 67.3 |
| Symnets (Zhang et al., 2019a) | ✗ | 47.7 | 72.9 | 78.5 | 64.2 | 71.3 | 74.2 | 64.2 | 48.8 | 79.5 | 74.5 | 52.6 | 82.7 | 67.6 |
| MDD (Zhang et al., 2019b) | ✗ | 54.9 | 73.7 | 77.8 | 60.0 | 71.4 | 71.8 | 61.2 | 53.6 | 78.1 | 72.5 | **60.2** | 82.3 | 68.1 |
| TADA (Wang et al., 2019a) | ✗ | 53.1 | 72.3 | 77.2 | 59.1 | 71.2 | 72.1 | 59.7 | 53.1 | 78.4 | 72.4 | 60.0 | 82.9 | 67.6 |
| BNM (Cui et al., 2020) | ✗ | 52.3 | 73.9 | 80.0 | 63.3 | 72.9 | 74.9 | 61.7 | 49.5 | 79.7 | 70.5 | 53.6 | 82.2 | 67.9 |
| BDG (Yang et al., 2020) | ✗ | 51.5 | 73.4 | 78.7 | 65.3 | 71.5 | 73.7 | 65.1 | 49.7 | 81.1 | 74.6 | 55.1 | 84.8 | 68.7 |
| SRDC (Tang et al., 2020) | ✗ | 52.3 | 76.3 | 81.0 | **69.5** | 76.2 | 78.0 | **68.7** | 53.8 | 81.7 | **76.3** | 57.1 | 85.0 | 71.3 |
| RSDA-MSTN (Gu et al., 2020) | ✗ | 53.2 | 77.7 | 81.3 | 66.4 | 74.0 | 76.5 | 67.9 | 53.0 | 82.0 | 75.8 | 57.8 | **85.4** | 70.9 |
| Source Only | ✓ | 44.6 | 67.3 | 74.8 | 52.7 | 62.7 | 64.8 | 53.0 | 40.6 | 73.2 | 65.3 | 45.4 | 78.0 | 60.2 |
| **+ELR** | ✓ | _52.4_ | _73.5_ | _77.3_ | _62.5_ | _70.6_ | _71.0_ | _61.1_ | _50.8_ | _78.9_ | _71.7_ | _56.7_ | _81.6_ | _67.3_ |
| SHOT (Liang et al., 2020) | ✓ | 57.1 | 78.1 | 81.5 | 68.0 | 78.2 | 78.1 | 67.4 | 54.9 | 82.2 | 73.3 | 58.8 | 84.3 | 71.8 |
| **+ELR** | ✓ | **58.7** | **78.9** | **82.1** | _68.5_ | _79.0_ | 77.5 | 68.2 | _57.1_ | 81.9 | _74.2_ | _59.5_ | 84.9 | **72.6** |
| G-SFDA (Yang et al., 2021b) | ✓ | 55.8 | 77.1 | 80.5 | 66.4 | 74.9 | 77.3 | 66.5 | 53.9 | 80.8 | 72.4 | 59.7 | 83.2 | 70.7 |
| **+ELR** | ✓ | _56.4_ | _77.6_ | _81.1_ | _67.1_ | _75.2_ | _77.9_ | 65.9 | _55.0_ | _81.2_ | 72.1 | _60.0_ | _83.6_ | _71.1_ |
| NRC (Yang et al., 2021a) | ✓ | 56.3 | 77.6 | 81.0 | 65.3 | 78.3 | 77.5 | 64.5 | 56.0 | 82.4 | 70.0 | 57.1 | 82.9 | 70.8 |
| **+ELR** | ✓ | _58.4_ | _78.7_ | _81.5_ | _69.2_ | **79.5** | **79.3** | 66.3 | **58.0** | _82.6_ | _73.4_ | _59.8_ | 85.1 | **72.6** |

that PCL (Zhang et al., 2021) leverages ETP at every epoch, so it cannot capture the benefits of ETP and is inappropriate for unbounded label noise due to the fast memorization speed in SFDA. As a comparison, we choose ELR since it leverages ETP at every batch. The blue lines show that leveraging ETP via ELR can address the memorization of noisy labels in SFDA.

# 5 EXPERIMENTS

We aim to improve the efficiency of existing SFDA algorithms by using ELR to leverage ETP. We evaluate the performance on four different SFDA benchmark datasets: Office-31 (Saenko et al., 2010), Office-Home (Venkateswara et al., 2017), VisDA (Peng et al., 2017) and DomainNet (Peng et al., 2019). Due to the limited space, the results on the dataset Office-31 and additional experimental details are provided in Appendix G.

**Evaluation.** We incorporate ELR into three existing baseline methods: SHOT (Liang et al., 2020), G-SFDA (Zhang & Sabuncu, 2018), and NRC (Yang et al., 2021a). SHOT uses k-means clustering and mutual information maximization strategy to train the representation network while freezing the final linear layer. G-SFDA aims to cluster target data with similar neighbors and attempts to maintain the source domain performance. NRC also explores the neighbors of target data by graph-based methods. ELR can be easily embedded into these methods by simply adding the regularization term into the loss function to optimize without affecting existing SFDA frameworks. We average the results based on three random runs.

**Results.** Tables 1-4 show the results before/after leveraging the early-time training phenomenon, where Table 4 is shown in Appendix G. Among these tables, the top part shows the results of conventional UDA methods, and the bottom part shows the results of SFDA methods. In the tables, we use SF to indicate whether the method is source free or not. We use Source Only + ELR to indicate ELR with self-training. The results show that ELR itself can boost the performances. As existing SFDA methods are not able to address unbounded label noise, incorporating ELR into these SFDA methods can further boost the performance. The four datasets, including all 31 pairs (e.g., $A \rightarrow D$) of tasks, show better performance after solving the unbounded label noise problem using the early-time training phenomenon. Meanwhile, solving the unbounded label noise on existing SFDA methods achieves state-of-the-art on all benchmark datasets. These SFDA methods also outperform most methods that need to access source data.

**Analysis about hyperparameters $\beta$ and $\lambda$.** The hyperparameter $\beta$ is chosen from {0.5, 0.6, 0.7, 0.8, 0.9, 0.99}, and $\lambda$ is chosen from {1, 3, 7, 12, 25}. We conduct the sensitivity study on hyperparameters of ELR on the DomainNet dataset, which is shown in Figure 3(a-b). In each Figure, the study is conducted by fixing the other hyperparameter to the optimal one. The performance is robust to the hyperparameter $\beta$ except $\beta = 0.99$. When $\beta = 0.99$, classifiers are sensitive to changes in learning curves. Thus, the performance degrades since the learning curves change quickly in the unbounded label noise scenarios. Meanwhile, the performance is also robust to the hyperparameter $\lambda$ except when $\lambda$ becomes too large. The hyperparameter $\lambda$ is to balance the effects of existing SFDA

algorithms and the effects of ELR. As we indicated in Tables 1-4, barely using ELR to address the SFDA problem is not comparable to these SFDA methods. Hence, a large value of $\lambda$ makes neural networks neglect the effects of these SFDA methods, leading to degraded performance.

Table 2: Accuracies (%) on DomainNet for ResNet50-based methods.

| Method | SF | R→C | R→P | R→S | C→R | C→P | C→S | P→R | P→C | P→S | S→R | S→C | S→P | **Avg** |
|---|---|---|---|---|---|---|---|---|---|---|---|---|---|---|
| MCD (Saito et al., 2018b) | ✗ | 61.9 | 69.3 | 56.2 | 79.7 | 56.6 | 53.6 | 83.3 | 58.3 | 60.9 | 81.7 | 56.2 | 66.7 | 65.4 |
| DANN (Ganin et al., 2016) | ✗ | 63.4 | 73.6 | 72.6 | 86.5 | 65.7 | 70.6 | 86.9 | 73.2 | 70.2 | 85.7 | 75.2 | 70.0 | 74.5 |
| DAN (Long et al., 2015) | ✗ | 64.3 | 70.6 | 58.4 | 79.4 | 56.7 | 60.0 | 84.5 | 61.6 | 62.2 | 79.7 | 65.0 | 62.0 | 67.0 |
| COAL (Tan et al., 2020) | ✗ | 73.9 | 75.4 | 70.5 | 89.6 | 70.0 | 71.3 | 89.8 | 68.0 | 70.5 | 88.0 | 73.2 | 70.5 | 75.9 |
| MDD (Zhang et al., 2019b) | ✗ | 77.6 | 75.7 | **74.2** | 89.5 | 74.2 | **75.6** | 90.2 | 76.0 | **74.6** | 86.7 | 72.9 | 73.2 | 78.4 |
| Source Only | ✓ | 53.7 | 71.6 | 52.9 | 70.8 | 49.5 | 58.3 | 85.2 | 59.6 | 59.1 | 30.6 | 74.8 | 65.7 | 61.0 |
| **+ELR** | ✓ | 70.2 | 81.7 | 61.7 | 79.9 | 63.8 | 67.0 | 90.0 | 72.1 | 66.8 | 85.1 | 78.5 | 68.8 | 73.8 |
| SHOT (Liang et al., 2020) | ✓ | 73.3 | 80.1 | 65.8 | **91.4** | 74.3 | 69.2 | 91.9 | 77.0 | 66.2 | 87.4 | 81.3 | 75.0 | 77.7 |
| **+ELR** | ✓ | **78.0** | 81.9 | 67.4 | 91.1 | **75.9** | 71.0 | 92.6 | 79.3 | 68.0 | 88.7 | 84.8 | 77.0 | **79.7** |
| G-SFDA (Yang et al., 2021b) | ✓ | 65.8 | 78.9 | 60.2 | 80.5 | 64.7 | 64.6 | 89.3 | 69.9 | 63.6 | 86.4 | 78.8 | 71.1 | 72.8 |
| **+ELR** | ✓ | 69.4 | 80.9 | 60.6 | 81.3 | 67.2 | 66.4 | 90.2 | 73.2 | 64.9 | 87.6 | 82.1 | 71.0 | 74.6 |
| NRC (Yang et al., 2021a) | ✓ | 69.8 | 81.1 | 62.9 | 83.4 | 74.4 | 66.3 | 90.3 | 73.4 | 65.2 | 88.2 | 82.2 | 75.8 | 76.4 |
| **+ELR** | ✓ | 75.6 | **82.2** | 65.7 | 91.2 | 77.2 | 68.5 | **92.7** | **79.8** | 67.5 | **89.3** | **85.1** | **77.6** | 79.4 |

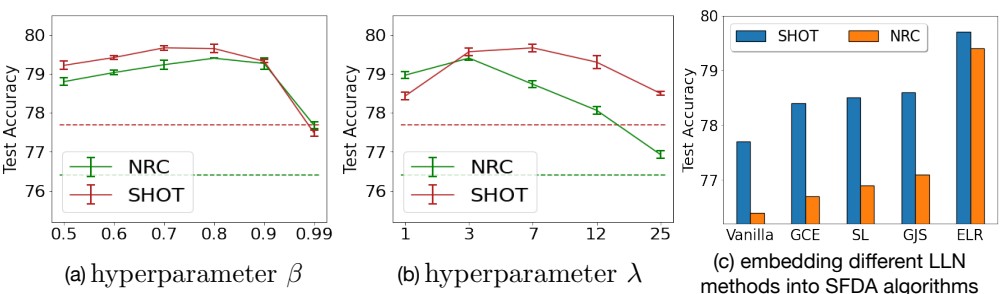

Figure 3: (a)-(b) show the test accuracy on the DomainNet dataset with respect to hyperparameters of ELR. (c) shows the test accuracy of incorporating various existing LLN methods into the SFDA methods on the DomainNet dataset.

## 5.1 DISCUSSION ON EXISTING LLN METHODS

As we formulate the SFDA as the problem of LLN, it is of interest to discuss some existing LLN methods. We mainly discuss existing LLN methods that can be easily embedded into the current SFDA algorithms. Based on this principle, we choose GCE (Zhang & Sabuncu, 2018), SL (Wang et al., 2019b) and GJS (Englesson & Azizpour, 2021) that have been theoretically proved to be robust to symmetric and asymmetric label noise, which are bounded label noise. We highlight that a more recent method GJS outperforms ELR in real-world noisy datasets. However, we will show that GJS is inferior to ELR in SFDA scenarios, because the underlying assumption for GJS does not hold in SFDA. Besides ELR, which leverages ETP, PCL is another method to leverage the same phenomenon, but we will show that it is also inappropriate for SFDA.

To show the effects of the existing LLN methods under the unbounded label noise, we test these LLN methods on various SFDA datasets with target data whose labels are generated by source models. As shown in Figure 4, GCE, SL, GJS, and PCL are better than CE but still not comparable to ELR. Our analysis indicates that ELR follows the principle of ETP, which is theoretically justified in SFDA scenarios by our Theorem 3.1. Methods GCE, SL, and GJS follow the bounded label noise assumption, which does not hold in SFDA. Hence, they perform worse than ELR in SFDA, even though GJS outperforms ELR in conventional LLN scenarios. PCL (Zhang et al., 2021) utilizes ETP to purify noisy labels of target data, but it performs significantly worse than ELR. As the memorization speed of the unbounded label noise is very fast, and classifiers memorize noisy labels within a few iterations (shown in Figure 2), purifying noisy labels every epoch is inappropriate for SFDA. However, we notice that PCL performs relatively better on DomainNet than on other datasets. The reason behind it is that the memorization speed in the DomainNet dataset is relatively slow than

Table 3: Accuracies (%) on VisDA-C (Synthesis → Real) for ResNet101-based methods.

| Method | SF | plane | bcycl | bus | car | horse | knife | mcycl | person | plant | sktbrd | train | truck | **Per-class** |
|---|---|---|---|---|---|---|---|---|---|---|---|---|---|---|
| DANN (Ganin et al., 2016) | ✗ | 81.9 | 77.7 | 82.8 | 44.3 | 81.2 | 29.5 | 65.1 | 28.6 | 51.9 | 54.6 | 82.8 | 7.8 | 57.4 |
| DAN (Long et al., 2015) | ✗ | 87.1 | 63.0 | 76.5 | 42.0 | 90.3 | 42.9 | 85.9 | 53.1 | 49.7 | 36.3 | 85.8 | 20.7 | 61.1 |
| ADR (Saito et al., 2018a) | ✗ | 94.2 | 48.5 | 84.0 | 72.9 | 90.1 | 74.2 | 92.6 | 72.5 | 80.8 | 61.8 | 82.2 | 28.8 | 73.5 |
| CDAN (Long et al., 2018) | ✗ | 85.2 | 66.9 | 83.0 | 50.8 | 84.2 | 74.9 | 88.1 | 74.5 | 83.4 | 76.0 | 81.9 | 38.0 | 73.9 |
| SAFN (Xu et al., 2019a) | ✗ | 93.6 | 61.3 | 84.1 | 70.6 | 94.1 | 79.0 | 91.8 | 79.6 | 89.9 | 55.6 | 89.0 | 24.4 | 76.1 |
| SWD (Lee et al., 2019) | ✗ | 90.8 | 82.5 | 81.7 | 70.5 | 91.7 | 69.5 | 86.3 | 77.5 | 87.4 | 63.6 | 85.6 | 29.2 | 76.4 |
| MDD (Zhang et al., 2019b) | ✗ | - | - | - | - | - | - | - | - | - | - | - | - | 74.6 |
| MCC (Jin et al., 2020) | ✗ | 88.7 | 80.3 | 80.5 | 71.5 | 90.1 | 93.2 | 85.0 | 71.6 | 89.4 | 73.8 | 85.0 | 36.9 | 78.8 |
| STAR (Lu et al., 2020) | ✗ | 95.0 | 84.0 | 84.6 | 73.0 | 91.6 | 91.8 | 85.9 | 78.4 | 94.4 | 84.7 | 87.0 | 42.2 | 82.7 |
| RWOT (Xu et al., 2020) | ✗ | 95.1 | 80.3 | 83.7 | **90.0** | 92.4 | 68.0 | 92.5 | 82.2 | 87.9 | 78.4 | **90.4** | **68.2** | 84.0 |
| Source Only | ✓ | 60.9 | 21.6 | 50.9 | 67.6 | 65.8 | 6.3 | 82.2 | 23.2 | 57.3 | 30.6 | 84.6 | 8.0 | 46.6 |
| **+ELR** | ✓ | 95.4 | 45.7 | 89.7 | 69.8 | 94.1 | 97.1 | **92.9** | 80.1 | 89.7 | 52.8 | 83.3 | 4.3 | 74.6 |
| SHOT (Liang et al., 2020) | ✓ | 94.3 | 88.5 | 80.1 | 57.3 | 93.1 | 94.9 | 80.7 | 80.3 | 91.5 | 89.1 | 86.3 | 58.2 | 82.9 |
| **+ELR** | ✓ | 95.8 | 84.1 | 83.3 | 67.9 | 93.9 | **97.6** | 89.2 | 80.1 | 90.6 | 90.4 | 87.2 | 48.2 | 84.1 |
| G-SFDA (Yang et al., 2021b) | ✓ | 96.0 | 87.6 | 85.3 | 72.8 | 95.9 | 94.7 | 88.4 | 79.0 | 92.7 | 93.9 | 87.2 | 43.7 | 84.8 |
| **+ELR** | ✓ | **97.3** | 89.1 | **89.8** | 79.2 | **96.9** | 97.5 | 92.2 | **82.5** | 95.8 | 94.5 | 87.3 | 34.5 | **86.4** |
| NRC (Yang et al., 2021a) | ✓ | 96.9 | 89.7 | 84.0 | 59.8 | 95.9 | 96.6 | 86.5 | 80.9 | 92.8 | 92.6 | 90.2 | 60.2 | 85.4 |
| **+ELR** | ✓ | 97.1 | **89.7** | 82.7 | 62.0 | 96.2 | 97.0 | 87.6 | 81.2 | 93.7 | 94.1 | 90.2 | 58.6 | 85.8 |

(i) Office-31  (ii) Office-Home  (iii) VisDA  (iv) DomainNet

Figure 4: Evaluation of label noise methods on SFDA problems. We use source models as an initialization of classifiers trained on target data and also use source models to annotate unlabeled target data. Then we treat the target datasets as noisy datasets and use different label noise methods to solve the memorization issue.

other datasets, which is shown in Figure 2. In conventional LLN scenarios, PCL does not suffer from the issue since the memorization speed is much lower than the conventional LLN scenarios.

In Figure 3(c), we also evaluate the performance by incorporating the existing LLN methods into the SFDA algorithms SHOT and NRC. Since PCL and SHOT assign pseudo labels to target data, PCL is incompatible with some existing SFDA methods and cannot be easily embedded into some SFDA algorithms. Hence, we only embed GCE, SL, GJS, and ELR into the SFDA algorithms. The figure illustrates that ELR still performs better than other LLN methods when incorporated into SHOT and NRC. We also notice that GCE, SL, and GJS provide marginal improvement to the vanilla SHOT and NRC methods. We think the label noise in SFDA datasets is the hybrid noise that consists of both bounded label noise and unbounded label noise due to the non-linearity of neural networks. The GCE, SL, and GJS can address the bounded label noise, while ELR can address both bounded and unbounded label noise. Therefore, these experiments demonstrate that using ELR to leverage ETP can successfully address the unbounded label noise in SFDA.

## 6 CONCLUSION

In this paper, we study SFDA from a new perspective of LLN by theoretically showing that SFDA can be viewed as the problem of LLN with the unbounded label noise. Under this assumption, we rigorously justify that robust loss functions are not able to address the memorization issues of unbounded label noise. Meanwhile, based on this assumption, we further theoretically and empirically analyze the learning behavior of models during the early-time training stage and find that ETP can benifit the SFDA problems. Through extensive experiments across multiple datasets, we show that ETP can be exploited by ELR to improve prediction performance, and it can also be used to enhance existing SFDA algorithms.

ACKNOWLEDGMENTS

This work is supported by Natural Sciences and Engineering Research Council of Canada (NSERC), Discovery Grants program.

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

# A    NEIGHBORS LABEL NOISE OBSERVATIONS ON REAL-WORLD DATASETS

This section provides more observed results and explanations of Neighbors' label noise during the Source-Free Domain Adaptation process on real-world datasets.

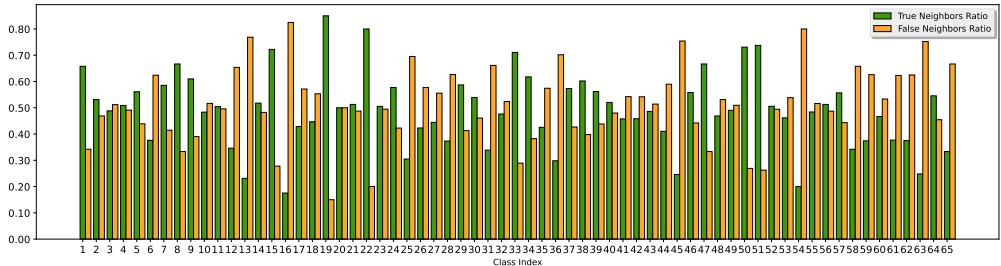

Figure 5: True/False Neighbors on Office-Home

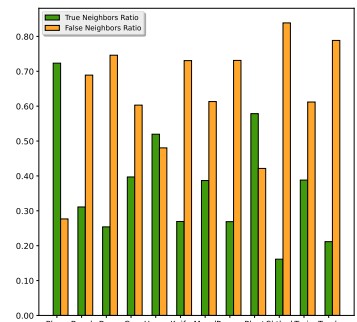

Figure 6: True/False Neighbors on VisDA

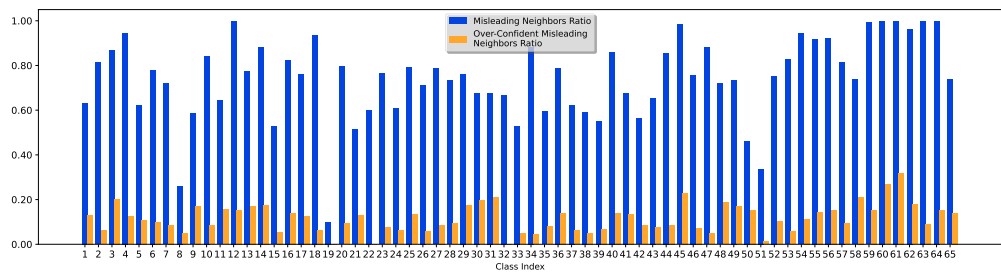

Figure 7: Neighbors Label Noise Analysis on Office-Home

Currently, most SFDA methods inevitably leverage the pseudo-labels for self-supervised learning or to learn the cluster structure of the target data in the feature space, in order to realize the domain adaptation goal. However, the pseudo labels generated by the source domain are usually noisy and of poor quality due to the domain distribution shift. Some neighborhood-based heuristic methods (Yang et al., 2021a;b) have been proposed to purify these target domain pseudo labels, which use the pseudo label of neighbors in the feature space to correct and reassign the central data's pseudo label. In fact, such methods rely on a strong assumption: a relatively high quality of the neighbors' pseudo label. However, in our experimental observations, we find that at the very beginning of the adaptation process, the similarity of two data points in the feature space can not fully represent their label space's connection. Furthermore, such methods are easy to provide useless and noisy prediction information for the central data. We will show some statistical results on VisDA and Office-Home, these two real-world datasets.

Following the neighborhood construction method in Yang et al. (2021a;b), we use the pre-trained source model to infer the target data, extract the feature space outputs and get the prediction results. We use the cosine similarity on the feature space to find the top $k$ similar neighbors (e.g., $k = 2$) for each data point (named as the central data point). Then, we collect the neighbors regarding the ground truth label of central data points and study the neighbor's quality for each class.

**Neighbors who do not belong to the correct category** We define the neighbors who do not belong to the same category as its central data point as *False Neighbor*, which means their ground-truth labels are not the same: $Y_{neighbor} \neq Y_{central}$. And the results of VisDA (train → validation) and Office-Home (Pr → Cl) datasets are shown in Figure 6 and Figure 5.

**Neighbors who can not provide useful prediction information** We further study the prediction information provided by such neighbors. Regardless of their true category properties, we consider neighbors whose *Predicted Label* is the same as the *Ground Truth Label* of the central data point to be *Useful Neighbors*; otherwise, they are *Misleading Neighbors*, as they can not provide the expected useful prediction information. We denote the *Misleading Neighbors Ratio* as the proportion of noisy neighbors among all neighbors for each class. Besides, as some methods heuristically utilize the predicted logits as the predicted probability or confidence score in the pseudo label purification process, we further study the *Over-Confident Misleading Neighbors Ratio* for each class. We defined the over-confident misleading neighbors ratio as the number of over-confident misleading neighbors (misleading neighbors with a high predicted logit, larger than 0.75) divided by the number of all neighbors per class. The results on VisDA and Office-Home are shown in Figure 1ii and Figure 7.

We want to clarify that the above exploratory experiment results can only reflect the phenomenon of unbounded noise in SFDA to some extent: the set of over-confidence misleading neighbors is non-empty can correspond, to some extent, to the fact that R is non-empty proved in Theorem 3.1; but the definition of misleading neighbors does not rigorously satisfies the definition of unbounded label noise.

## B RELATIONSHIP BETWEEN MISLABELING ERROR AND DOMAIN SHIFT

In this part, we focus on explaining the relationship between the label noise and the domain shift, as illustrated in Figure 9. The following theorem characterizes the relationship between the labeling error and the domain shift.

**Theorem B.1.** *Without loss of generality, we assume that the $\boldsymbol{\Delta}$ is positively correlated with the vector $\boldsymbol{\mu_2} - \boldsymbol{\mu_1}$, i.e., $\boldsymbol{\Delta}^\top(\boldsymbol{\mu_2} - \boldsymbol{\mu_1}) > 0$. Let $f_S$ be the Bayes optimal classifier for the source domain $S$. Then*

$$\Pr_{(\mathbf{x},y)\sim\mathcal{D}_T}[f_S(\mathbf{x}) \neq y] = \frac{1}{2}\Phi(-\frac{d_1}{\sigma}) + \frac{1}{2}\Phi(-\frac{d_2}{\sigma}), \qquad (6)$$

*where $d_1 = \left\|\frac{\boldsymbol{\mu_2}-\boldsymbol{\mu_1}}{2} - \mathbf{c}\right\| \operatorname{sign}(\left\|\frac{\boldsymbol{\mu_2}-\boldsymbol{\mu_1}}{2}\right\| - \|\mathbf{c}\|)$, $d_2 = \left\|\frac{\boldsymbol{\mu_2}-\boldsymbol{\mu_1}}{2} + \mathbf{c}\right\|$, $\mathbf{c} = (\boldsymbol{\mu_2} - \boldsymbol{\mu_1})\frac{\boldsymbol{\Delta}^\top(\boldsymbol{\mu_2}-\boldsymbol{\mu_1})}{\|\boldsymbol{\mu_2}-\boldsymbol{\mu_1}\|^2}$, and $\Phi$ is the standard normal cumulative distribution function.*

Theorem B.1 indicates that the labeling error for the target domain can be represented by a function of the domain shift $\boldsymbol{\Delta}$, which can be shown numerically in Figure 8. The projection of the domain shift $\boldsymbol{\Delta}$ on the vector $\boldsymbol{\mu_2} - \boldsymbol{\mu_1}$ is given by $\mathbf{c}$. Since $\mathbf{c}$ is on the direction of $\boldsymbol{\mu_2} - \boldsymbol{\mu_1}$, $\mathbf{c}$ can also be represented by $\alpha(\boldsymbol{\mu_2} - \boldsymbol{\mu_1})$, where $\alpha \in \mathbb{R}$ characterizes the magnitude of the domain shift. More specifically, in Figure 8, we present the relationship between the mislabeling rate and $\alpha$ for all possible $\boldsymbol{\Delta}$. When $\boldsymbol{\Delta}$ is positively correlated with $\boldsymbol{\mu_2} - \boldsymbol{\mu_1}$ (assumption in Theorem B.1), we have $\alpha > 0$, and when $\boldsymbol{\Delta}$ is negatively correlated with $\boldsymbol{\mu_2} - \boldsymbol{\mu_1}$, we obtain $\alpha < 0$. In both situations, we can observe that the labeling error increases with the absolute value of $\alpha$ increasing, which implies that the more severe the domain shift is, the greater the mislabeling error will be obtained. Besides, we note that when the source and target domains are the same, the mislabeling error in Eq. (6) is minimized and degraded to the Bayes error, which cannot be reduced (Fukunaga, 2013). This corresponds to the situation when $\boldsymbol{\Delta}$ is perpendicular to $\boldsymbol{\mu_2} - \boldsymbol{\mu_1}$, $\mathbf{c} = \mathbf{0}$, and $\alpha = 0$ shown in Figure 8.

### B.1 PROOFS FOR THEOREM B.1

*Proof.* The Bayes classifier $f_S$ predicts $\mathbf{x}$ to the first component when

$$\log \frac{\Pr[y=1|X=\mathbf{x}]}{\Pr[y=-1|X=\mathbf{x}]} > 0. \qquad (7)$$

Since the distributions of the two components with the same priors for the source domain are given by $\mathcal{N}(\boldsymbol{\mu_1}, \sigma^2\mathbf{I}_d)$ and $\mathcal{N}(\boldsymbol{\mu_2}, \sigma^2\mathbf{I}_d)$, respectively. Based on Bayes' rule, Eq. (7) is equivalent to

$$\log \frac{\Pr[X=\mathbf{x}|y=1]}{\Pr[X=\mathbf{x}|y=-1]} > 0 \qquad (8)$$

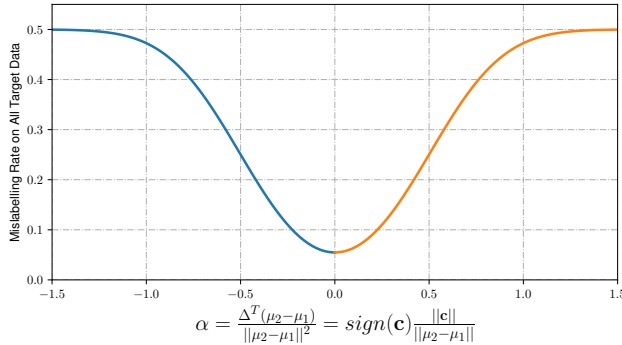

Figure 8: Plot of Mislabeling Rate with different $\alpha$. We define $\mathbf{c}$ as the projection of the domain shift $\boldsymbol{\Delta}$ on the vector $\boldsymbol{\mu_2} - \boldsymbol{\mu_1}$, and $\alpha$ represents the magnitude of domain shift projected on $\boldsymbol{\mu_2} - \boldsymbol{\mu_1}$.

Solving the left hand side of Eq. (8) by using the knowledge of two multivariate Gaussian distributions, we get

$$h_S(\mathbf{x}) := \log \frac{\Pr[X = \mathbf{x}|y = 1]}{\Pr[X = \mathbf{x}|y = -1]} = \frac{\mathbf{x}^\top(\boldsymbol{\mu_1} - \boldsymbol{\mu_2})}{\sigma^2} - \frac{\|\boldsymbol{\mu_1}\|^2 - \|\boldsymbol{\mu_2}\|^2}{2\sigma^2}. \tag{9}$$

So $f_S$ predicts $\mathbf{x}$ to the first component when $h_S(\mathbf{x}) > 0$ and $f_S$ predicts $\mathbf{x}$ to the second component when $h_S(\mathbf{x}) \leq 0$ The decision boundary is $\mathbf{z}$ such that $h_S(\mathbf{z}) = 0$. When there is no domain shift $\boldsymbol{\Delta} = \mathbf{0}$, we have $\mathcal{D}_S = \mathcal{D}_T$, and the mislabeling rate is the Bayes error, which is given by:

$$\Pr_{(\mathbf{x},y)\sim\mathcal{D}_S}[f_S(\mathbf{x}) \neq y] = \frac{1}{2} \Pr_{\mathbf{x}\sim\mathcal{N}(\boldsymbol{\mu_1},\sigma^2\mathbf{I}_d)}[h_S(\mathbf{x}) < 0|y = 1] + \frac{1}{2} \Pr_{\mathbf{x}\sim\mathcal{N}(\boldsymbol{\mu_2},\sigma^2\mathbf{I}_d)}[h_S(\mathbf{x}) > 0|y = -1] \tag{10}$$

We first study the first term in Eq. (10):

$$\Pr_{\mathbf{x}\sim\mathcal{N}(\boldsymbol{\mu_1},\sigma^2\mathbf{I}_d)}[h_S(\mathbf{x}) < 0|y = 1]$$

$$= \int \cdots \int_{\{\mathbf{x}|\mathbf{x}^\top(\boldsymbol{\mu_1}-\boldsymbol{\mu_2})<\frac{\|\boldsymbol{\mu_1}\|^2-\|\boldsymbol{\mu_2}\|^2}{2}\}} \frac{1}{(2\pi\sigma^2)^{\frac{d}{2}}} \exp\left(-\frac{\|\mathbf{x} - \boldsymbol{\mu_1}\|^2}{2\sigma^2}\right) dx_1 dx_2 \cdots dx_d$$

$$= \int \cdots \int_{\{\mathbf{x}|-\infty<x_1,x_2,\ldots,x_{d-1}<\infty,d_0<x_d\}} \frac{1}{(2\pi\sigma^2)^{\frac{d}{2}}} \exp\left(-\frac{\sum_{i=1}^d x_i^2}{2\sigma^2}\right) dx_1 dx_2 \cdots dx_d$$

$$= \int_{d_0}^{\infty} \frac{1}{2\pi\sigma^2} \exp\left(-\frac{x_d^2}{2\sigma^2}\right) dx_d$$

$$= \Phi(-\frac{d_0}{\sigma}),$$

where the second equality is because of the rotationally symmetric property for isotropic Gaussian random vectors, $\Phi$ is the cumulative distribution function of the standard Gaussian distribution, and $d_0 = \|(\boldsymbol{\mu_2} - \boldsymbol{\mu_1})/2\|$. Applying the similar mathematical steps for the second term in Eq. (10), and take them into Eq. (10):

$$\Pr_{(\mathbf{x},y)\sim\mathcal{D}_S}[f_S(\mathbf{x}) \neq y] = \Phi(-\frac{\|\boldsymbol{\mu_2} - \boldsymbol{\mu_1}\|}{2\sigma}). \tag{11}$$

When there is no domain shift, the labeling error is the Bayes error, which is expressed by Eq. (11).

Then we consider the case when $\boldsymbol{\Delta} \neq \mathbf{0}$. The distributions of the first and the second component are $\mathcal{N}(\boldsymbol{\mu_1} + \boldsymbol{\Delta}, \sigma^2\mathbf{I}_d)$ and $\mathcal{N}(\boldsymbol{\mu_2} + \boldsymbol{\Delta}, \sigma^2\mathbf{I}_d)$, respectively. Notice that the decision boundary $\mathbf{z}$ is the affine hyperplane. Any shift paralleled to this affine hyperplane will not affect the final component predictions. The domain shift $\boldsymbol{\Delta}$ can be decomposed into the sum of two vectors: the one is paralleled to this affine hyperplane, and another is perpendicular to the hyperplane. It is straightforward to

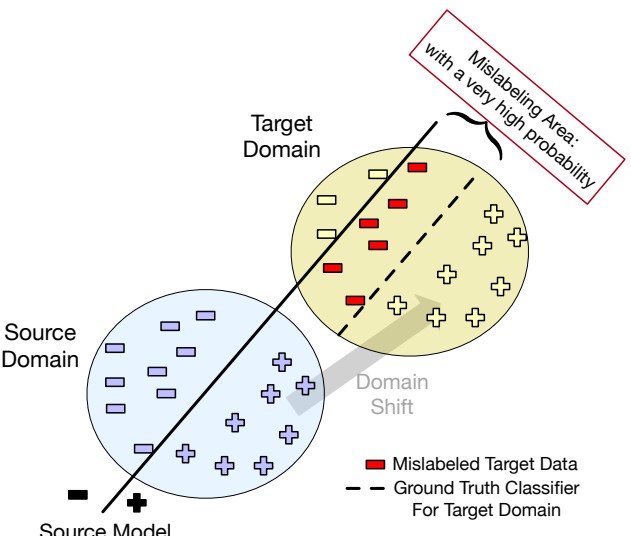

Figure 9: Illustration of noisy labels generated by the domain shift.

verify that $\boldsymbol{\mu_2} - \boldsymbol{\mu_1}$ is perpendicular to the hyperplane. Thus, we project the domain shift $\boldsymbol{\Delta}$ onto the vector $\boldsymbol{\mu_2} - \boldsymbol{\mu_1}$ to get the component of $\boldsymbol{\Delta}$ that is perpendicular to the hyperplane, which is given by:

$$\mathbf{c} = (\boldsymbol{\mu_2} - \boldsymbol{\mu_1}) \frac{\boldsymbol{\Delta}^\top (\boldsymbol{\mu_2} - \boldsymbol{\mu_1})}{\|\boldsymbol{\mu_2} - \boldsymbol{\mu_1}\|^2}. \tag{12}$$

Since we assume $\boldsymbol{\Delta}$ is positively correlated to the vector $\boldsymbol{\mu_2} - \boldsymbol{\mu_1}$, $\alpha = \frac{\boldsymbol{\Delta}^\top (\boldsymbol{\mu_2} - \boldsymbol{\mu_1})}{\|\boldsymbol{\mu_2} - \boldsymbol{\mu_1}\|^2}$ can be regarded as the magnitude of the domain shift along the direction $\boldsymbol{\mu_2} - \boldsymbol{\mu_1}$. Note that the results also hold for the case where $\boldsymbol{\Delta}$ is negatively correlated to $\boldsymbol{\mu_2} - \boldsymbol{\mu_1}$. The whole proof can be obtained by following the very similar proof steps for the positively correlated case.

The mislabeling rate of the optimal source classifier $f_S$ on target data is:

$$\Pr_{(\mathbf{x},y)\sim\mathcal{D}_T}[f_S(\mathbf{x}) \neq y] = \frac{1}{2} \Pr_{\mathcal{N}(\boldsymbol{\mu_1}+\boldsymbol{\Delta},\sigma^2\mathbf{I}_d)}[h_S(\mathbf{x}) < 0|y=1] + \frac{1}{2} \Pr_{\mathcal{N}(\boldsymbol{\mu_2}+\boldsymbol{\Delta},\sigma^2\mathbf{I}_d)}[h_S(\mathbf{x}) > 0|y=-1] \tag{13}$$

We first calculate the first term of Eq. (13). Following the same tricks discussed above:

$$\Pr_{\mathbf{x}\sim\mathcal{N}(\boldsymbol{\mu_1}+\boldsymbol{\Delta},\sigma^2\mathbf{I}_d)}[h_S(\mathbf{x}) < 0|y=1]$$

$$= \Pr_{\mathbf{x}\sim\mathcal{N}(\boldsymbol{\mu_1}+\mathbf{c},\sigma^2\mathbf{I}_d)}[h_S(\mathbf{x}) < 0|y=1]$$

$$= \int\cdots\int_{\{\mathbf{x}|\mathbf{x}^\top(\boldsymbol{\mu_1}-\boldsymbol{\mu_2})<\frac{\|\boldsymbol{\mu_1}\|^2-\|\boldsymbol{\mu_2}\|^2}{2}\}} \frac{1}{(2\pi\sigma^2)^{\frac{d}{2}}} \exp\left(-\frac{\|\mathbf{x} - \boldsymbol{\mu_1} - \boldsymbol{\Delta}\|^2}{2\sigma^2}\right) dx_1 dx_2 \cdots dx_d$$

$$= \int\cdots\int_{\{\mathbf{x}|-\infty<x_1,x_2,\ldots,x_{d-1}<\infty,d_1<x_d\}} \frac{1}{(2\pi\sigma^2)^{\frac{d}{2}}} \exp\left(-\frac{\sum_{i=1}^d x_i^2}{2\sigma^2}\right) dx_1 dx_2 \cdots dx_d$$

$$= \int_{d_1}^{\infty} \frac{1}{2\pi\sigma^2} \exp\left(-\frac{x_d^2}{2\sigma^2}\right) dx_d$$

$$= \Phi(-\frac{d_1}{\sigma}), \tag{14}$$

where $d_1 = \left\|\frac{\boldsymbol{\mu_2}-\boldsymbol{\mu_1}}{2} - \mathbf{c}\right\| \operatorname{sign}(\left\|\frac{\boldsymbol{\mu_2}-\boldsymbol{\mu_1}}{2}\right\| - \|\mathbf{c}\|)$.

Similarly, the second term is given by:

$$\Pr_{\mathbf{x}\sim\mathcal{N}(\boldsymbol{\mu_2}+\boldsymbol{\Delta},\sigma^2\mathbf{I}_d)}[h_S(\mathbf{x})>0|y=-1]=\Phi(-\frac{d_2}{\sigma}), \tag{15}$$

where $d_2=\left\|\frac{\boldsymbol{\mu_2}-\boldsymbol{\mu_1}}{2}+\mathbf{c}\right\|$.

Taking Eq. (14) and Eq. (15) into Eq. (13), we have

$$\Pr_{(\mathbf{x},y)\sim\mathcal{D}_T}[f_S(\mathbf{x})\neq y]=\frac{1}{2}\Phi(-\frac{d_1}{\sigma})+\frac{1}{2}\Phi(-\frac{d_2}{\sigma}). \tag{16}$$

$\square$

## C  PROOFS FOR THEOREM 3.1

*Proof.* Without loss of generality, we choose to assume $\boldsymbol{\mu_2}=\boldsymbol{\mu_1}+\sigma\mathbf{1}_d$ as the convenient way to present our results. From the proof for Theorem B.1, we know that $\mathbf{x_0}=\frac{\boldsymbol{\mu_1}+\boldsymbol{\mu_2}}{2}+\boldsymbol{\Delta}$ is at the decision boundary such that $h_T(\mathbf{x_0})=0$, where

$$h_T(\mathbf{x})=\frac{\mathbf{x}^\top(\boldsymbol{\mu_1}-\boldsymbol{\mu_2})}{\sigma^2}-\frac{\|\boldsymbol{\mu_1}+\boldsymbol{\Delta}\|^2-\|\boldsymbol{\mu_2}+\boldsymbol{\Delta}\|^2}{2\sigma^2}.$$

Let $f_T$ be the optimal Bayes classifier for the target domain, which can be obtained the same way as $f_S$ mentioned in B.1. The equation $h_T(\mathbf{x_0})=0$ implies that

$$\Pr_{(\mathbf{x},y)\sim\mathcal{D}_T}[y=1|X=\mathbf{x_0}]=\Pr_{(\mathbf{x},y)\sim\mathcal{D}_T}[y=-1|X=\mathbf{x_0}].$$

Note that $\mathbf{x_0}$ is on the affine hyperplane $\mathbf{z}$ where $h_T(\mathbf{z})=0$. Any data points on this hyperplane will have the equal probabilities to be correctly classified. We start from this hyperplane and calculate another point $\mathbf{x_1}$, where $\Pr_{(\mathbf{x},y)\sim\mathcal{D}_T}[y=1|X=\mathbf{x_1}]$ is at least $\frac{1-\delta}{\delta}\Pr_{(\mathbf{x},y)\sim\mathcal{D}_T}[y=-1|X=\mathbf{x_1}]$. Thus, for any points that are mislabeled and far away from $\mathbf{x_1}$ will result in $\Pr_{(\mathbf{x},y)\sim\mathcal{D}_T}[y=1|X=\mathbf{x_1}]\geq 1-\delta$. We first aim to find such a data point $\mathbf{x_1}$. Let $\mathbf{x_1}=\mathbf{x_0}-m_0\sigma\mathbf{1}_d$, where $m_0$ is the scalar measures the distance between the point $\mathbf{x_1}$ to the hyperplane $\mathbf{z}$. We need to find $m_0$ such that

$$\frac{P_T(\mathbf{x_1}|y=1)}{P_T(\mathbf{x_1}|y=-1)}\geq 1-\delta, \tag{17}$$

where

$$\frac{P_T(\mathbf{x_1}|y=1)}{P_T(\mathbf{x_1}|y=-1)}$$
$$=\exp\left(-\frac{\|\mathbf{x_1}-\boldsymbol{\mu_1}-\boldsymbol{\Delta}\|^2}{2\sigma^2}+\frac{\|\mathbf{x_1}-\boldsymbol{\mu_2}-\boldsymbol{\Delta}\|^2}{2\sigma^2}\right)$$
$$=\exp\left(-\frac{\left\|\frac{\boldsymbol{\mu_2}-\boldsymbol{\mu_1}}{2}-m_0\sigma\mathbf{1}_d\right\|^2}{2\sigma^2}+\frac{\left\|\frac{\boldsymbol{\mu_2}-\boldsymbol{\mu_1}}{2}+m_0\sigma\mathbf{1}_d\right\|^2}{2\sigma^2}\right)$$
$$=\exp\left(m_0d\right) \tag{18}$$

Taking Eq. (18) into Eq. (17), we get $m_0\geq(\log\frac{1-\delta}{\delta})/d$. Since the isotropic Gaussian random vectors has the rotationally symmetric property, we can transform the integration of multivariate normal distribution to standard normal distribution with different intervals of integration. Then any data points from a region that have at most $\|\mathbf{x_1}-\boldsymbol{\mu_1}-\boldsymbol{\Delta}\|$ distance to its mean $\boldsymbol{\mu_1}+\boldsymbol{\Delta}$ will have at least $0.99$ probability coming from the first component. Let the region $\mathbf{R}_1$ be:

$$\mathbf{R}_1=\{\mathbf{x}:\|\mathbf{x}-\boldsymbol{\mu_1}-\boldsymbol{\Delta}\|\leq\|\mathbf{x_1}-\boldsymbol{\mu_1}-\boldsymbol{\Delta}\|\}$$

Equivalently, taking $\mathbf{R}_1$ can be simplified:

$$\mathbf{R}_1=\{\mathbf{x}:\|\mathbf{x}-\boldsymbol{\mu_1}-\boldsymbol{\Delta}\|\leq\sigma(\frac{\sqrt{d}}{2}-\frac{\log\frac{1-\delta}{\delta}}{\sqrt{d}})\}$$

The region $\mathbf{R}_1$ is valid when data dimension $d$ is large. This is realistic in practice. Since neural networks are usually dealing with high dimension data, for example $d \gg (1)$, the region $\mathbf{R}_1$ is valid.

On the other hand, we aim to find a region $\mathbf{R}_2$ where all data points are mislabeled. From the proof for Theorem 1, the source classifier $h_S$ is given by

$$h_S(\mathbf{x}) = \frac{\mathbf{x}^\top (\boldsymbol{\mu_1} - \boldsymbol{\mu_2})}{\sigma^2} - \frac{\|\boldsymbol{\mu_1}\|^2 - \|\boldsymbol{\mu_2}\|^2}{2\sigma^2}. \tag{19}$$

Any data points are classified to the second component if $h_S(\mathbf{x}) < 0$. Hence

$$\mathbf{R}_2 = \{\mathbf{x} : \mathbf{x}^\top \mathbf{1}_d > \frac{\sigma d + 2\boldsymbol{\mu}_1^\top \mathbf{1}_d}{2}\}$$

We take the intersection of $\mathbf{R}_1$ and $\mathbf{R}_2$, all data points from this intersection are (1) having at least $1 - \delta$ probability coming from the first component, and (2) being classified to the second component. Formally, for $(\mathbf{x}, y) \sim \mathcal{D}_T$, if $\mathbf{x} \in \mathbf{R}_1 \bigcap \mathbf{R}_2$, then

$$\Pr[f_S(\mathbf{x}) \neq y] \geq 1 - \delta, \tag{20}$$

We note that $\mathbf{x} \in \mathbf{R}_1 \bigcap \mathbf{R}_2$ is non-empty when $(\log \frac{1-\delta}{\delta})/d < \alpha$, where $\alpha = \frac{\boldsymbol{\Delta}^\top (\boldsymbol{\mu_2} - \boldsymbol{\mu_1})}{\|\boldsymbol{\mu_2} - \boldsymbol{\mu_1}\|^2}$ is the magnitude of the domain shift along with the direction $\boldsymbol{\mu_2} - \boldsymbol{\mu_1}$. Since $\mathbf{x}_1$ is chosen from $\mathbf{R}_1$, to verify that $\mathbf{R}_1 \bigcap \mathbf{R}_2$ is non-empty, we only need to verify that $\mathbf{x}_1$ also belongs to $\mathbf{R}_2$.

$\mathbf{x}_1 \in \mathbf{R}_2$ if and only if:

$$\mathbf{x}_1^\top \mathbf{1}_d > \frac{\sigma d + 2\boldsymbol{\mu}_1^\top \mathbf{1}_d}{2}$$
$$(\boldsymbol{\mu}_1 + \mathbf{c} + \frac{\sigma}{2}\mathbf{1}_d - m_0 \sigma \mathbf{1}_d)^\top \mathbf{1}_d > \frac{\sigma d + 2\boldsymbol{\mu}_1^\top \mathbf{1}_d}{2}$$
$$(\boldsymbol{\mu}_1 + \alpha\sigma\mathbf{1}_d + \frac{\sigma}{2}\mathbf{1}_d - m_0 \sigma \mathbf{1}_d)^\top \mathbf{1}_d > \frac{\sigma d + 2\boldsymbol{\mu}_1^\top \mathbf{1}_d}{2}$$
$$(\alpha - m_0)\sigma d > 0,$$

where $\mathbf{c} = (\boldsymbol{\mu_2} - \boldsymbol{\mu_1})\frac{\boldsymbol{\Delta}^\top (\boldsymbol{\mu_2} - \boldsymbol{\mu_1})}{\|\boldsymbol{\mu_2} - \boldsymbol{\mu_1}\|^2}$.

Therefore, if $\alpha > m_0 \geq (\log \frac{1-\delta}{\delta})/d$, $\mathbf{R}_1 \bigcap \mathbf{R}_2$ is non-empty.

Next, we show $\Pr_{(\mathbf{x},y)\sim\mathcal{D}_T}[\mathbf{x} \in \mathbf{R}]$ increases as $\alpha$ increases.

Let event $\mathbf{A}_0$ be a set of $\mathbf{x}$ such that they are mislabeled by $f_S$ (i.e. $f_S(\mathbf{x}) \neq y$). Let event $\mathbf{A}_1$ be a set of $\mathbf{x}$ such that they are from the first component but are mislabeled to the second component with a probability $\Pr[f_S(\mathbf{x} \neq y)] < 1 - \delta$. Let event $\mathbf{A}_2$ be a set of $\mathbf{x}$ such that they are from the second component but are mislabeled to the first component with a probability $\Pr[f_S(\mathbf{x} \neq y)] < 1 - \delta$. Thus

$$\Pr_{(\mathbf{x},y)\sim\mathcal{D}_T}[\mathbf{x} \in \mathbf{R}] = \Pr_{(\mathbf{x},y)\sim\mathcal{D}_T}[\mathbf{A}_0] - \Pr_{(\mathbf{x},y)\sim\mathcal{D}_T}[\mathbf{A}_1] - \Pr_{(\mathbf{x},y)\sim\mathcal{D}_T}[\mathbf{A}_2] \tag{21}$$

Let event $\mathbf{A}_3$ be a set of $\mathbf{x}$ such that they are from the first component such that $\Pr[f_S(\mathbf{x} \neq y)] < 1-\delta$ or $\Pr[f_S(\mathbf{x} = y)] < 1 - \delta$. Let event $\mathbf{A}_4$ be a set of $\mathbf{x}$ such that they are from the second component but are mislabeled to the first component. For $\Pr[\mathbf{A}_3]$,

$$\Pr_{(\mathbf{x},y)\sim\mathcal{N}(\boldsymbol{\mu_1}+\boldsymbol{\Delta},\sigma^2\mathbf{I}_d)}[\mathbf{A}_3] = \Pr_{(\mathbf{x},y)\sim\mathcal{N}(\boldsymbol{\mu_1}+\boldsymbol{\Delta},\sigma^2\mathbf{I}_d)}[\mathbf{R}_1^{\complement}],$$

which does not change as the domain shift $\boldsymbol{\Delta}$ varies. Meanwhile,

$$\Pr_{(\mathbf{x},y)\sim\mathcal{N}(\boldsymbol{\mu_2}+\boldsymbol{\Delta},\sigma^2\mathbf{I}_d)}[\mathbf{A}_4] = \Phi(-\frac{\left\|\frac{\boldsymbol{\mu_2}-\boldsymbol{\mu_1}}{2} + \mathbf{c}\right\|}{\sigma}),$$

which is given by Eq. (15). By our assumption, the domain shift $\boldsymbol{\Delta}$ is positively correlated with the vector $\boldsymbol{\mu_2} - \boldsymbol{\mu_1}$. So when $\alpha$ increases, $\Pr_{(\mathbf{x},y)\sim\mathcal{N}(\boldsymbol{\mu_2}+\boldsymbol{\Delta},\sigma^2\mathbf{I}_d)}[\mathbf{A}_4]$ decreases.

Since $\mathbf{A}_1 \subseteq \mathbf{A}_3$ and $\mathbf{A}_2\mathbf{A}_4$, the probability measure on $\mathbf{R}$ is given by:

$$\Pr_{(\mathbf{x},y)\sim\mathcal{D}_T}[\mathbf{x} \in \mathbf{R}] = \Pr_{(\mathbf{x},y)\sim\mathcal{D}_T}[\mathbf{A}_0] - \Pr_{(\mathbf{x},y)\sim\mathcal{D}_T}[\mathbf{A}_1] - \Pr_{(\mathbf{x},y)\sim\mathcal{D}_T}[\mathbf{A}_2]$$

$$\geq \Pr_{(\mathbf{x},y)\sim\mathcal{D}_T}[\mathbf{A}_0] - \Pr_{(\mathbf{x},y)\sim\mathcal{D}_T}[\mathbf{A}_3] - \Pr_{(\mathbf{x},y)\sim\mathcal{D}_T}[\mathbf{A}_4], \qquad (22)$$

where the first term is the mislabeling rate that increases as $\alpha$ increases (given by Theorem B.1); the second term is a constant; the third term decreases as as $\alpha$ increases. The equality in Eq. (22) holds when $\alpha \to \infty$. Therefore, when the magnitude of the domain shift $\alpha$ increases, the lower bound of $\Pr_{(\mathbf{x},y)\sim\mathcal{D}_T}[\mathbf{x} \in \mathbf{R}]$ increases, which forces more points to break the conventional LLN assumption.

$\square$

## D  BACKGROUND INTRODUCTION AND PROOFS FOR LEMMA 3.2

Learning with label noise is an important task and topic in deep learning and modern artificial intelligence research. The main idea behind it is robust training, which can be further divided into fine-grained categories, such as robust architecture, robust regularization, robust loss design, and simple selection (Song et al., 2022). For example, for the robust architecture-based methods, they propose to modify the deep model's architecture, including adding an adaptation layer or leveraging a dedicated module, to learn the label transition process and to tackle the noisy label. In addition, the robust regularization approaches usually enforce the DNN to overfit less to false-labeled examples by adopting a regularizer, explicitly or implicitly. For instance, Yi et al. (2022) proposed to utilize a contrastive regularization term to learn a noisy-label robust representation. Recently, with the widespread implementation of AI technologies, the topics of trustworthiness and fairness have drawn a lot of interest (Liu et al., 2019; Shui et al., 2022a;b). How to provide trustworthy and fair learning in LLN problems is a significant research direction. In this paper, we will, however, develop our discussion based on the robust loss methods in LLN.

In this section, we will first introduce the concepts and technical details of some noise-robust loss based LLN methods, including GCE (Zhang & Sabuncu, 2018), SL (Wang et al., 2019b), NCE (Ma et al., 2020), and GJS (Englesson & Azizpour, 2021). Then, we will present the proof details of Lemma 3.2.

### D.1  NOISE-ROBUST LOSS FUNCTIONS IN LLN METHODS

Among the numerous studies of LLN methods, loss correction is a major branch of research. The main idea of loss correction is to modify the loss function and make it robust to noisy labels.

As indicated in Ma et al. (2020), the loss function $\ell$ is defined to be noise robust if $\sum_{k=1}^{K} \ell(h(\mathbf{x}), k) = C$, where $C$ is a positive constant and $K$ is the overall class number of label space. For example, the most widely utilized Cross-Entropy (CE) loss is unbounded and therefore is not robust to the label noise. Some LLN studies show that existing loss functions such as mean absolute error (MAE) (Ghosh et al., 2017), reverse cross entropy (RCE) (Wang et al., 2019b), normalized cross entropy (NCE) (Ma et al., 2020), and normalized focal loss (NFL) are noise-robust and that combining them with CE can help mitigate the sensitivity of the model to noisy labels.

More specifically, for a given data $(\mathbf{x}, \mathbf{y})$ and a classifier $h(\mathbf{x})$, GCE (Zhang & Sabuncu, 2018) leverages the negative Box-Cox transformation as a loss function, which can exploit the benefits of both the noise-robustness provided by MAE and the implicit weighting scheme of CE:

$$\ell_{\mathbf{GCE}}(h(\mathbf{x}), \mathbf{e}_k) = \frac{(1 - h_k(\mathbf{x})^q)}{q}$$

where $q \in (0, 1]$ is a hyperparameter to be decided.

Another noise-robust loss based method SL (Wang et al., 2019b) proposes combining the reverse cross entropy (RCE) loss, which is noise tolerant, with CE loss and obtain the $\mathcal{L}_{\mathbf{SL}}$:

$$\ell_{\mathbf{SL}} = \alpha\ell_{\mathbf{CE}} + \beta\ell_{\mathbf{RCE}}$$

$$= -(\alpha \sum_{k=1}^{K} q(k|\mathbf{x})logp(k|\mathbf{x}) + \beta \sum_{k=1}^{K} p(k|\mathbf{x})logq(k|\mathbf{x}))$$

where $p(k|\mathbf{x})$ is the predicted distribution over labels by classifier $h(\mathbf{x})$ and $q(k|\mathbf{x})$ is the ground truth class distribution conditioned on sample $\mathbf{x}$.

GJS (Zhang & Sabuncu, 2018) utilizes the multi-distribution generalization of Jensen-Shannon Divergence as loss function, which has been proven noise-robust and is in fact a generalization of CE and MAE. Concretely, the generalized JS divergence and GJS loss are defined as:

$$D_{\mathbf{GJS}_\pi} = \sum_{i=1}^{M} \pi_i D_{\mathbf{KL}}\Big(\boldsymbol{p}^{(i)}\Big|\Big|\sum_{j=1}^{M} \pi_j \boldsymbol{p}^{(j)}\Big)$$

$$\ell_{\mathbf{GJS}}(\mathbf{x}, \mathbf{y}, h) = \frac{D_{\mathbf{GJS}_\pi}(\mathbf{y}, h(\tilde{\mathbf{x}}^{(2)}), ..., h(\tilde{\mathbf{x}}^{(M)}))}{Z}$$

where $\boldsymbol{\pi}$, $\boldsymbol{p}^{(i)}$ are categorical distributions over $K$ classes, $\tilde{\mathbf{x}}^{(i)} \sim \mathcal{A}(\mathbf{x})$, a random perturbation of sample $\mathbf{x}$, and $Z = -(1 - \pi_1)\log(1 - \pi_1)$

Further, Ma et al. (2020) shows a simple loss normalization scheme which can be applied for any loss $\mathcal{L}$:

$$\ell_{\mathbf{NORM}} = \frac{\ell(h(\mathbf{x}), \mathbf{y})}{\sum_{k=1}^{K} \ell(h(\mathbf{x}), k)}$$

The study found that the normalized loss can indeed satisfy the robustness condition. However, it will also cause an underfitting problem in some situations.

Note that generalized cross entropy (GCE (Zhang & Sabuncu, 2018)) extends MAE and symmetric loss (SL (Wang et al., 2019b)) extends RCE. So we study GCE and SL in our experiments instead studying MAE and RCE. Besides, GJS (Englesson & Azizpour, 2021) is shown to be tightly bounded around $\sum_{k=1}^{K} \ell(h(\mathbf{x}), k)$. All these methods have shown to be noise tolerant under either bounded random label noise or bounded class-conditional label noise with additional assumption that $R(h^\star) = 0$. We show that under the same assumption with unbounded label noise datasets, these methods are not noise tolerant in section D.2.

### D.2 PROOFS FOR LEMMA 3.2

*Proof.* Let $\eta_{yk}(\mathbf{x})$ be the $\Pr[\tilde{Y} = k|Y = y, X = \mathbf{x}]$ probability of observing a noisy label $k$ given the ground-truth label $y$ and a sample $\mathbf{x}$. Let $\eta_y(\mathbf{x}) = \sum_{k \neq y} \eta_{yk}(\mathbf{x})$. The risk of $h$ under noisy data is given by

$$
\begin{aligned}
\widetilde{R}(h) =& \mathbb{E}_{\mathbf{x}, \tilde{y}}[\ell_{\mathrm{LLN}}(h(\mathbf{x}), \tilde{y})] \\
=& \mathbb{E}_{\mathbf{x}} \mathbb{E}_{y|\mathbf{x}} \mathbb{E}_{\tilde{y}|\mathbf{x}, y}[\ell_{\mathrm{LLN}}(h(\mathbf{x}), \tilde{y})] \\
=& \mathbb{E}_{\mathbf{x}, y}\Big[(1 - \eta_y(\mathbf{x}))\ell_{\mathrm{LLN}}(h(\mathbf{x}), y) + \sum_{k \neq y} \eta_{yk}(\mathbf{x})\ell_{\mathrm{LLN}}(h(\mathbf{x}), k)\Big] \\
=& \mathbb{E}_{\mathbf{x}, y}\Big[(1 - \eta_y(\mathbf{x}))\big(\sum_{k=1}^{K} \ell_{\mathrm{LLN}}(h(\mathbf{x}), k) - \sum_{k \neq y} \ell_{\mathrm{LLN}}(h(\mathbf{x}), k)\big) + \sum_{k \neq y} \eta_{yk}(\mathbf{x})\ell_{\mathrm{LLN}}(h(\mathbf{x}), k)\Big] \\
=& \mathbb{E}_{\mathbf{x}, y}\Big[(1 - \eta_y(\mathbf{x}))\big(C - \sum_{k \neq y} \ell_{\mathrm{LLN}}(h(\mathbf{x}), k)\big) + \sum_{k \neq y} \eta_{yk}(\mathbf{x})\ell_{\mathrm{LLN}}(h(\mathbf{x}), k)\Big] \\
=& \mathbb{E}_{\mathbf{x}, y}\Big[(1 - \eta_y(\mathbf{x}))C\Big] - \mathbb{E}_{\mathbf{x}, y}\Big[\sum_{k \neq y}\big(1 - \eta_y(\mathbf{x}) - \eta_{yk}(\mathbf{x})\big)\ell_{\mathrm{LLN}}(h(\mathbf{x}), k)\Big].
\end{aligned}
\tag{23}
$$

Since Eq. (23) holds for both $\tilde{h}^\star$ and $h^\star$, we have

$$\widetilde{R}(\tilde{h}^\star) = \mathbb{E}_{\mathbf{x}, y}\Big[(1 - \eta_y(\mathbf{x}))C\Big] - \mathbb{E}_{\mathbf{x}, y}\Big[\sum_{k \neq y}\big(1 - \eta_y(\mathbf{x}) - \eta_{yk}(\mathbf{x})\big)\ell_{\mathrm{LLN}}(\tilde{h}^\star(\mathbf{x}), k)\Big] \tag{24}$$

and

$$\widetilde{R}(h^\star) = \mathbb{E}_{\mathbf{x}, y}\Big[(1 - \eta_y(\mathbf{x}))C\Big] - \mathbb{E}_{\mathbf{x}, y}\Big[\sum_{k \neq y}\big(1 - \eta_y(\mathbf{x}) - \eta_{yk}(\mathbf{x})\big)\ell_{\mathrm{LLN}}(h^\star(\mathbf{x}), k)\Big]. \tag{25}$$

As $\tilde{h}^\star$ is the minimizer of $\widetilde{R}(h)$, $\widetilde{R}(\tilde{h}^\star) \leq \widetilde{R}(h^\star)$. Then we combine Eq. (24) and Eq. (25), we have

$$\mathbb{E}_{\mathbf{x},y}\left[\sum_{k\neq y}\left(1 - \eta_y(\mathbf{x}) - \eta_{yk}(\mathbf{x})\right)\left(\ell_{\text{LLN}}(h^\star(\mathbf{x}),k) - \ell_{\text{LLN}}(\tilde{h}^\star(\mathbf{x}),k)\right)\right] \leq 0. \tag{26}$$

We note that $\ell_{\text{LLN}}(\tilde{h}^\star(\mathbf{x}),k) \geq \ell_{\text{LLN}}(h^\star(\mathbf{x}),k)$ implies $p_k(\mathbf{x}) = 0$ and $p_y(\mathbf{x}) = 1$ for $k \neq y$, where $p_k(\mathbf{x})$ is the probability output by $\tilde{h}^\star$ for predicting the sample $\mathbf{x}$ to be the class $k$. This argument is proved given by Wang et al. (2019b); Ghosh et al. (2017); Yang et al. (2021b); Ma et al. (2020) (Theorem 1&2 in Ghosh et al. (2017), Theorem 1 in Wang et al. (2019b), Lemma 1&2 in Ma et al. (2020) and Theorem 1&2 in Englesson & Azizpour (2021)).

To let $\ell_{\text{LLN}}(\tilde{h}^\star(\mathbf{x}),k) \geq \ell_{\text{LLN}}(h^\star(\mathbf{x}),k)$ holds for all inputs $\mathbf{x}$, previous studies assume the bounded label noise, which is given by

$$1 - \eta_y(\mathbf{x}) - \eta_{yk}(\mathbf{x}) > 0 \ \forall\mathbf{x} \ \text{s.t.} \ P(X = \mathbf{x}) > 0. \tag{27}$$

For random label noise which assumes that the mislabeling probability from the ground-truth label to any other label is the same for all inputs, i.e. $\eta_{ji}(\mathbf{x}) = a_0 \ \forall i \neq j$, where $a_0$ is a constant. Let $\eta = (K-1)a_0$, then Eq. (27) is degraded to

$$1 - \eta - \frac{\eta}{K-1} > 0$$
$$1 > \frac{K}{K-1}\eta$$
$$\eta < 1 - \frac{1}{K}.$$

This bounded assumption is commonly assumed by Wang et al. (2019b); Ghosh et al. (2017); Yang et al. (2021b); Ma et al. (2020) (Theorem 1 in Ghosh et al. (2017), Theorem 1 in Wang et al. (2019b), Lemma 1 in Ma et al. (2020) and Theorem 1 in Englesson & Azizpour (2021)).

For class-conditional label noise, which assumes the $\eta_{ji}(\mathbf{x}_1) = \eta_{ji}(\mathbf{x}_2)$ for any inputs $\mathbf{x}_1$ and $\mathbf{x}_2$. Let $\eta_{ji}(\mathbf{x}) = \eta_{ji}$, Then the bounded assumption Eq. (27) is degraded to

$$\eta_{yk} < 1 - \eta_y.$$

This bounded assumption is also commonly assumed, and it can be found in Theorem 2 in Ghosh et al. (2017), Theorem 1 in Wang et al. (2019b), 2 in Ma et al. (2020) and Theorem 2 in Englesson & Azizpour (2021).

However, in SFDA, we proved that the following event $\mathbf{B}$ holds with a probability at least $1 - \delta$:

$$1 - \eta_y(\mathbf{x}) - \eta_{yk}(\mathbf{x}) < 0 \ \forall\mathbf{x} \in \mathbf{R}. \tag{28}$$

Indeed, we first denote $\mathbf{B_1} = \{\tilde{y} \neq y | \mathbf{x} \in \mathbf{R}\}$ by the event that $\mathbf{x} \in \mathbf{R}$ is mislabeled. Then

$$\begin{aligned}
\Pr[\mathbf{B}] &= \Pr[\mathbf{B}|\mathbf{B_1}] + \Pr[\mathbf{B}|\mathbf{B_1^C}]\Pr[\mathbf{B_1^C}] \\
&\geq \Pr[\mathbf{B}|\mathbf{B_1}]\Pr[\mathbf{B_1}] \\
&\geq 1 - \delta
\end{aligned}$$

Given the result in Eq. (28), and combined it with the Eq. (26), we have

$$\ell_{\text{LLN}}(\tilde{h}^\star(\mathbf{x}),k) \leq \ell_{\text{LLN}}(h^\star(\mathbf{x}),k).$$

When the event $\mathbf{B}$ holds, the condition $\ell_{\text{LLN}}(\tilde{h}^\star(\mathbf{x}),k) \leq \ell_{\text{LLN}}(h^\star(\mathbf{x}),k)$ holds.

Note that only $\ell_{\text{LLN}}(\tilde{h}^\star(\mathbf{x}),k) \geq \ell_{\text{LLN}}(h^\star(\mathbf{x}),k)$ means $p_k(\mathbf{x}) = 0$ for $k \neq y$ and $p_y(\mathbf{x}) = 1$ for $k \neq y$. It means that the optimal classifier $\tilde{h}^\star$ from noisy data can make correct predictions on any inputs, which is consistent with the optimal classifier $h^\star$ obtained from clean data.

As for the condition $\ell_{\text{LLN}}(\tilde{h}^\star(\mathbf{x}),k) \leq \ell_{\text{LLN}}(h^\star(\mathbf{x}),k)$, we can get $p_k(\mathbf{x}) = 1$ for a $k \neq y$, which means that the optimal classifier $\tilde{h}^\star$ from noisy data cannot make correct predictions on samples

$\mathbf{x} \in \mathbf{R}$. To verify this, we use the robust loss function RCE $\ell_{\mathrm{RCE}}$ as an example, and it can be easily generalized to other robust los functions mentioned above. Based on the definition of the RCE loss (Wang et al., 2019b), we have

$$
\begin{aligned}
\ell_{\mathrm{RCE}}(\tilde{h}^{\star}(\mathbf{x}), k) =& C_{\mathrm{RCE}}(1 - p_k(\mathbf{x})) \\
\ell_{\mathrm{RCE}}(h^{\star}(\mathbf{x}), k) =& C_{\mathrm{RCE}},
\end{aligned}
$$

where $C_{\mathrm{RCE}} > 0$ is a constant. The above equations show that any $0 \le p_k(\mathbf{x}) \le 1$ can make the condition $\ell_{\mathrm{LLN}}(\tilde{h}^{\star}(\mathbf{x}), k) \le \ell_{\mathrm{LLN}}(h^{\star}(\mathbf{x}), k)$ hold. Meanwhile, $\tilde{h}^{\star}$ is the global minimizer of the risk over the noisy data, which makes $\tilde{h}^{\star}$ memorize the noisy dataset.

Therefore, $\tilde{h}^{\star}$ makes incorrect predictions for $\mathbf{x} \in \mathbf{R}$ such that $p_k(\mathbf{x}) = 1$ for a $k \ne y$, and $h^{\star}$ is the global optimal over clean data, which gives correct predictions for $\mathbf{x} \in \mathbf{R}$ such that $p_k(\mathbf{x}) = 1$ for a $k = y$. That completes the proof as $h^{\star}$ makes different predictions on $\mathbf{x} \in \mathbf{R}$ compared to $\tilde{h}^{\star}$.

$\square$

## E  PROOFS FOR THEOREM 4.1

The proof for Theorem 4.1 is partially adopted from Liu et al. (2020). Note that we are dealing with unbounded label noise, whereas the bounded label noise is considered in Liu et al. (2020). As indicated in Liu et al. (2020), $T$ is set as the smallest positive integer such that $\theta_T^{\top} \boldsymbol{\mu} \ge 0.1$, and $T = \Omega(1/\eta)$ with high probability. Parameters $\theta$ is initialized by Kaiming initialization (He et al., 2015) that $\theta_0 \sim \mathcal{N}(0, \frac{2}{d}\mathbf{I}_d)$, and $|\theta_0^{\top} \boldsymbol{\mu}|$ converges in probability to 0. For simplicity, we assume $\theta_0 = 0$ without loss of generality. The proof consists of two parts. The first part is to show that $\theta_{T-1}$ is highly positively correlated with the ground truth classifier. The second part is to show that the prediction accuracy on mislabeled samples can be represented as the correlation between the learned classifier and the ground truth classifier.

*Proof.* **To begin with, we show the first part.** Let samples $\mathbf{x}_i = y_i(\boldsymbol{\mu} - \sigma \mathbf{z}_i)$, where $\mathbf{z} \sim \mathcal{N}(0, \mathbf{I}_d)$. The gradient of the logistic loss function with respect to the parameter $\theta$ is given by:

$$
\begin{aligned}
\nabla_{\theta} \mathcal{L}(\theta_t) =& \frac{1}{2n} \sum_{i=1}^{n} \mathbf{x}_i\big(\tanh(\theta_t^{\top} \mathbf{x}_i) - \tilde{y}_i\big) \\
=& \underbrace{-\frac{1}{2n} \sum_{i=1}^{n} \tilde{y}_i \mathbf{x}_i}_{\text{①}} + \underbrace{\frac{1}{2n} \sum_{i=1}^{n} \mathbf{x}_i \tanh(\theta_t^{\top} \mathbf{x}_i)}_{\text{②}}
\end{aligned} \tag{29}
$$

Then we will show that $-\boldsymbol{\mu}^{\top} \nabla_{\theta} \mathcal{L}(\theta_t)$ is lower bounded by a positive number. We first show the bound on ① in Eq. (29). Since $\mathbf{x}_i$ is sampled from standard normal distribution, $\frac{1}{n} \sum_{i=1}^{n} \tilde{y}_i \boldsymbol{\mu}^{\top} \mathbf{x}_i$ has limited variance. By the law of large number, $\frac{1}{n} \sum_{i=1}^{n} \tilde{y}_i \boldsymbol{\mu}^{\top} \mathbf{x}_i$ converges in probability to its mean. Therefore,

$$
\begin{aligned}
\mathbb{E}[\tilde{y} \mathbf{x}^{\top} \boldsymbol{\mu}] =& \mathbb{E}[\tilde{y} \boldsymbol{\mu}^{\top} \mathbf{x} \mathbb{1}\{y\mathbf{x}^{\top} \boldsymbol{\mu} \le r\}] + \mathbb{E}[\tilde{y} \boldsymbol{\mu}^{\top} \mathbf{x} \mathbb{1}\{y\mathbf{x}^{\top} \boldsymbol{\mu} > r\}] \\
=& \mathbb{E}[\mathbb{E}[\tilde{y} \boldsymbol{\mu}^{\top} \mathbf{x} \mathbb{1}\{y\mathbf{x}^{\top} \boldsymbol{\mu} \le r\}]|y] \\
& + \mathbb{E}[\mathbb{E}[\tilde{y} \boldsymbol{\mu}^{\top} \mathbf{x} \mathbb{1}\{y\mathbf{x}^{\top} \boldsymbol{\mu} > r\}]|y] \\
=& \mathbb{E}[-\boldsymbol{\mu}^{\top} \mathbf{x} \mathbb{1}\{\mathbf{x}^{\top} \boldsymbol{\mu} \le r\}|y = 1] + \mathbb{E}[\boldsymbol{\mu}^{\top} \mathbf{x} \mathbb{1}\{\mathbf{x}^{\top} \boldsymbol{\mu} > r\}|y = 1]
\end{aligned}
$$

Note that $\mathbf{x}|y = 1$ is a Gaussian random vector with independent entries, we have $\mathbf{x}^\top \boldsymbol{\mu} \overset{\mathrm{d}}{=} w + 1$, where $w \sim \mathcal{N}(0, \sigma^2)$. Therefore, the above expectation is equivalent to

$$
\begin{aligned}
\mathbb{E}[\tilde{y}\mathbf{x}^\top \boldsymbol{\mu}] &= - \int_{-\infty}^{r-1} (w+1)\, \mathrm{d}\mathbb{P}_w + \int_{r-1}^{\infty} (w+1)\, \mathrm{d}\mathbb{P}_w \\
&= - \int_{-\infty}^{r-1} w\, \mathrm{d}\mathbb{P}_w + \int_{r-1}^{+\infty} w\, \mathrm{d}\mathbb{P}_w - \int_{-\infty}^{r-1} \mathrm{d}\mathbb{P}_w + \int_{r-1}^{+\infty} \mathrm{d}\mathbb{P}_w \\
&= \int_{r-1}^{1-r} \mathrm{d}\mathbb{P}_w - \int_{-\infty}^{r-1} w\, \mathrm{d}\mathbb{P}_w + \int_{r-1}^{+\infty} w\, \mathrm{d}\mathbb{P}_w \\
&= \mathrm{Erf}[\frac{1-r}{\sqrt{2}\sigma}] + 2\frac{\sigma}{\sqrt{2\pi}} \exp\big(-\frac{(r-1)^2}{2\sigma^2}\big),
\end{aligned}
\tag{30}
$$

where $\mathrm{Erf}[x] = \frac{2}{\sqrt{\pi}} \int_0^x e^{-t^2}\, \mathrm{d}t$. Note that $r < 1$, which means that most half of samples are mislabeled. Thus

$$
\frac{1}{2}\mathbb{E}[\tilde{y}_i \boldsymbol{\mu}^\top \mathbf{x}_i] = \frac{1}{2}\mathrm{Erf}[\frac{1-r}{\sqrt{2}\sigma}] + \frac{\sigma}{\sqrt{2\pi}} \exp\big(-\frac{(r-1)^2}{2\sigma^2}\big) > 0.
$$

Now we deal with the ②in in Eq. (29).

$$
\frac{1}{2n}|\boldsymbol{\mu}^\top \big(\sum_{i=1}^n \tanh(\theta_t^\top x_i)\big)| = \frac{1}{2n}|\mathbf{q}^\top \mathbf{p}|
$$

$$
\leq \frac{1}{2n}\|\mathbf{q}\|\,\|\mathbf{p}\|,
\tag{31}
$$

$\mathbf{q} = (\boldsymbol{\mu}^\top \mathbf{x}_1, \boldsymbol{\mu}^\top \mathbf{x}_2, \ldots, \boldsymbol{\mu}^\top \mathbf{x}_n) \in \mathbb{R}^n$, and $\mathbf{p} = (\tanh(\theta_t^\top x_1), \tanh(\theta_t^\top x_2), \ldots, \tanh(\theta_t^\top x_n)) \in \mathbb{R}^n$.

By triangle inequality of the norm,

$$
\|\mathbf{q}\| = \|\mathbf{q} - \mathbf{1} + \mathbf{1}\| \leq \|\mathbf{q} - \mathbf{1}\| + \|\mathbf{1}\| = \sqrt{n} + \|\mathbf{q} - \mathbf{1}\|,
$$

where $\mathbf{q} - \mathbf{1}$ is a random vector with Gaussian coordinates. By Lemma E.1,

$$
\|\mathbf{q} - \mathbf{1}\|/\sigma \leq 2\sigma\sqrt{n}
\tag{32}
$$

with probability $1 - \delta$ when $n \geq c_1 \log 1/\delta$, where $c_1$ is a constant.

On the other hand,

$$
\begin{aligned}
\big\|\mathbf{p} - \tanh(\theta_t^\top \boldsymbol{\mu})\mathbf{1}_n + \tanh(\theta_t^\top \boldsymbol{\mu})\mathbf{1}_n\big\| &\leq \big\|\tanh(\theta_t^\top \boldsymbol{\mu})\mathbf{1}_n\big\| + \big\|\mathbf{p} - \tanh(\theta_t^\top \boldsymbol{\mu})\mathbf{1}_n\big\| \\
&\leq \big\|\tanh(\theta_t^\top \boldsymbol{\mu})\mathbf{1}_n\big\| + \|\theta_t\|\,\|\mathbf{q} - 1\| \\
&= \tanh(\theta_t^\top \boldsymbol{\mu})\sqrt{n} + 2\sigma\sqrt{n}\,\|\theta_t\|,
\end{aligned}
\tag{33}
$$

where the second inequality is by Lemma 9 from Liu et al. (2020), the last inequality by Lemma E.1.

Then we take Eq. (31) and Eq.(33) together, and then take them and Eq.(30) into $-\boldsymbol{\mu}^\top \nabla_\theta \mathcal{L}(\theta_t)$, which gives us:

$$
-\nabla_\theta \mathcal{L}(\theta_t)^\top \boldsymbol{\mu} \geq \frac{1}{2}\mathrm{Erf}[\frac{1-r}{\sqrt{2}\sigma}] + \frac{\sigma}{\sqrt{2\pi}} \exp\big(-\frac{(r-1)^2}{2\sigma^2}\big) - \sigma(\tanh(\theta_t^\top \boldsymbol{\mu}) + 2\sigma\,\|\theta_t\|)
\tag{34}
$$

By Lemma 8 from Liu et al. (2020), we have $\sup_{\theta \in \mathbb{R}^d} \|\nabla_\theta \mathcal{L}(\theta)\| \leq 1 + 2\sigma$. Therefore, Eq. (34) can be rewritten as:

$$
\begin{aligned}
\frac{-\nabla_\theta \mathcal{L}(\theta_t)^\top \boldsymbol{\mu}}{\|\nabla_\theta \mathcal{L}(\theta_t)\|} &\geq \frac{\mathrm{Erf}[\frac{1-r}{\sqrt{2}\sigma}] + 2\frac{\sigma}{\sqrt{2\pi}} \exp\big(-\frac{(r-1)^2}{2\sigma^2}\big)}{1 + 2\sigma} - \frac{\sigma(\tanh(\theta_t^\top \boldsymbol{\mu}) + 2\sigma\,\|\theta_t\|)}{1 + 2\sigma} \\
&\geq \frac{b_0}{1 + 2\sigma} - \frac{\sigma(\tanh(\theta_t^\top \boldsymbol{\mu}) + 2\sigma\,\|\theta_t\|)}{1 + 2\sigma},
\end{aligned}
\tag{35}
$$

where we let $b_0 = \frac{1}{2}\text{Erf}[\frac{1-r}{\sqrt{2}\sigma}] + \frac{\sigma}{\sqrt{2\pi}}\exp\left(-\frac{(r-1)^2}{2\sigma^2}\right)$.

Then we prove $\frac{-\nabla_\theta \mathcal{L}(\theta_t)^\top \boldsymbol{\mu}}{\|\nabla_\theta \mathcal{L}(\theta_t)\|} \geq \frac{1}{10}\frac{b_0}{1+2\sigma}$ by mathematical induction, which can help us get rid of the dependence on $\theta_t$ for the lower bound in Eq. (35).

For $t = 0$, the inequality holds trivially. By the gradient descent algorithm, $\theta_{t+1} = -\eta \sum_{i=0}^{t} \nabla_\theta \mathcal{L}(\theta_i)$, where $-\boldsymbol{\mu}^\top \nabla_\theta \mathcal{L}(\theta_i)/\|\nabla_\theta \mathcal{L}(\theta_i)\| \geq \frac{1}{10}\frac{b_0}{1+2\sigma}$.

$$
\begin{aligned}
\frac{\theta_{t+1}^\top \boldsymbol{\mu}}{\|\theta_{t+1}\|} &\geq \frac{-\eta \sum_{i=0}^{t} \boldsymbol{\mu}^\top \nabla_\theta \mathcal{L}(\theta_i)}{\eta \left\|\sum_{i=0}^{t} \nabla_\theta \mathcal{L}(\theta_i)\right\|} \\
&\geq \frac{\frac{1}{10}\frac{b_0}{1+2\sigma}(\sum_{i=0}^{t} \|\nabla_\theta \mathcal{L}(\theta_i)\|)}{\sum_{i=0}^{t} \|\nabla_\theta \mathcal{L}(\theta_i)\|} \\
&\geq \frac{1}{10}\frac{b_0}{1+2\sigma}
\end{aligned}
$$

As $t+1 < T$, we have $\|\theta_{t+1}\| \leq 10\frac{1+2\sigma}{b_0}\theta_{t+1}^\top \boldsymbol{\mu} \leq \frac{1+2\sigma}{b_0}$. Taking it into Eq. (35), we have

$$
\frac{-\nabla_\theta \mathcal{L}(\theta_t)^\top \boldsymbol{\mu}}{\|\nabla_\theta \mathcal{L}(\theta_t)\|} \geq \frac{b_0}{1+2\sigma} - \frac{\sigma(0.1 + \frac{1+2\sigma}{b_0})}{1+2\sigma}
$$

To show $\frac{-\nabla_\theta \mathcal{L}(\theta_t)^\top \boldsymbol{\mu}}{\|\nabla_\theta \mathcal{L}(\theta_t)\|}$ is lower bounded by $\frac{1}{10}\frac{b_0}{1+2\sigma}$, we need to have

$$
h(\sigma) = \frac{9}{10}\frac{b_0}{1+2\sigma} - \sigma(0.1 + \frac{1+2\sigma}{b_0}) > 0
$$

It is straightforward to verify that $h(\sigma = 0) > 0$ and it can be verified that when $0 < \sigma < c_0$, we have $h'(\sigma) > 0$. Therefore, for $0 < \sigma < c_0$ and any $t < T-1$

$$
\frac{-\nabla_\theta \mathcal{L}(\theta_t)^\top \boldsymbol{\mu}}{\|\nabla_\theta \mathcal{L}(\theta_t)\|} \geq \frac{1}{10}\frac{b_0}{1+2\sigma}
$$

Hence by gradient descent algorithm $\theta_T = -\eta \sum_{i=0}^{T-1} \nabla_\theta \mathcal{L}(\theta_i)$ and the same proof above, we have

$$
\frac{\theta_T^\top \boldsymbol{\mu}}{\|\theta_T\|} \geq \frac{1}{10}\frac{b_0}{1+2\sigma} \tag{36}
$$

**For the second part:** the prediction accuracy on mislabeled sample set $B$ converges in probability to its mean. Therefore, the expectation of the prediction accuracy on mislabeled samples is given by

$$
\begin{aligned}
\mathbb{E}[\mathbb{1}\{\text{sign}(\theta_T^\top \mathbf{x}) = y\}] &= \mathbb{E}[\mathbb{1}\{\text{sign}(y\theta_T^\top(\boldsymbol{\mu} - \sigma\mathbf{z})) = y\}] \\
&= \mathbb{E}[\mathbb{1}\{\text{sign}(\theta_T^\top(\boldsymbol{\mu} - \sigma\mathbf{z})) = 1\}] \\
&= \Pr[\sigma\theta_T^\top \mathbf{z} > \theta_T^\top \boldsymbol{\mu}] \tag{37}
\end{aligned}
$$

Note that $\mathbf{z}$ is a standard Gaussian vector, $\theta_T^\top \mathbf{z}$ is distributed as $\mathcal{N}(0, \|\theta_T\|^2)$ Thus, Eq. (37) is equivalent to $\Phi(\frac{\theta_T^\top \boldsymbol{\mu}}{\sigma\|\theta_T\|})$.

By the inequality $1 - \Phi(x) \leq \exp\{-x^2/2\}$ for $x > 0$, then we have

$$
\Phi(\frac{\theta_T^\top \boldsymbol{\mu}}{\sigma\|\theta_T\|}) \geq 1 - \exp\{-\frac{(\frac{\theta_T^\top \boldsymbol{\mu}}{\sigma\|\theta_T\|})^2}{2}\} \geq 1 - \exp\{-\frac{1}{200}\left(\frac{b_0}{(1+2\sigma)\sigma}\right)^2\}
$$

We denote $g(\sigma)$ by:

$$
g(\sigma) = \frac{\text{Erf}[\frac{1-r}{\sqrt{2}\sigma}]}{2(1+2\sigma)\sigma} + \frac{\exp\left(-\frac{(r-1)^2}{2\sigma^2}\right)}{\sqrt{2\pi}(1+2\sigma)},
$$

where $g(\sigma) > 0$ for any $\sigma > 0$. Note that $g(\sigma) \to \infty$ when $\sigma \to 0$, and $g(\sigma)$ is monotone decreasing as $\sigma$ increases since $g'(\sigma) < 0$ for $\sigma > 0$.

$\square$

**Lemma E.1.** *Let $X = (X_1, X_2, \ldots, X_n) \in \mathbb{R}^n$ be a random vector with independent, Gaussian coordinates $X_i$ with $\mathbb{E}[X_i] = 0$ and $\mathbb{E}[X_i^2] = 1 < \infty$. Then*

$$\Pr[|\, \|X\|_2 - \sqrt{n}| \geq \sqrt{n}] \leq 2 \exp\left(-an\right),$$

*where $a > 0$ is a constant.*

*Proof.* The Gaussian concentration result is taken from Proposition 5.34 in Vershynin (2018), which will be used here for proving Theorem 4.1. □

## F  ADDITIONAL LEARNING CURVES

We provide additional learning curves on DomainNet dataset, shown in Figure 10. The dataset contains 12 pairs of tasks showing: (1) target classifiers have higher prediction accuracy during the early-training time; (2) leverage ETP by using ELR can alleviate the memorization of unbounded noisy labels generated by source models.

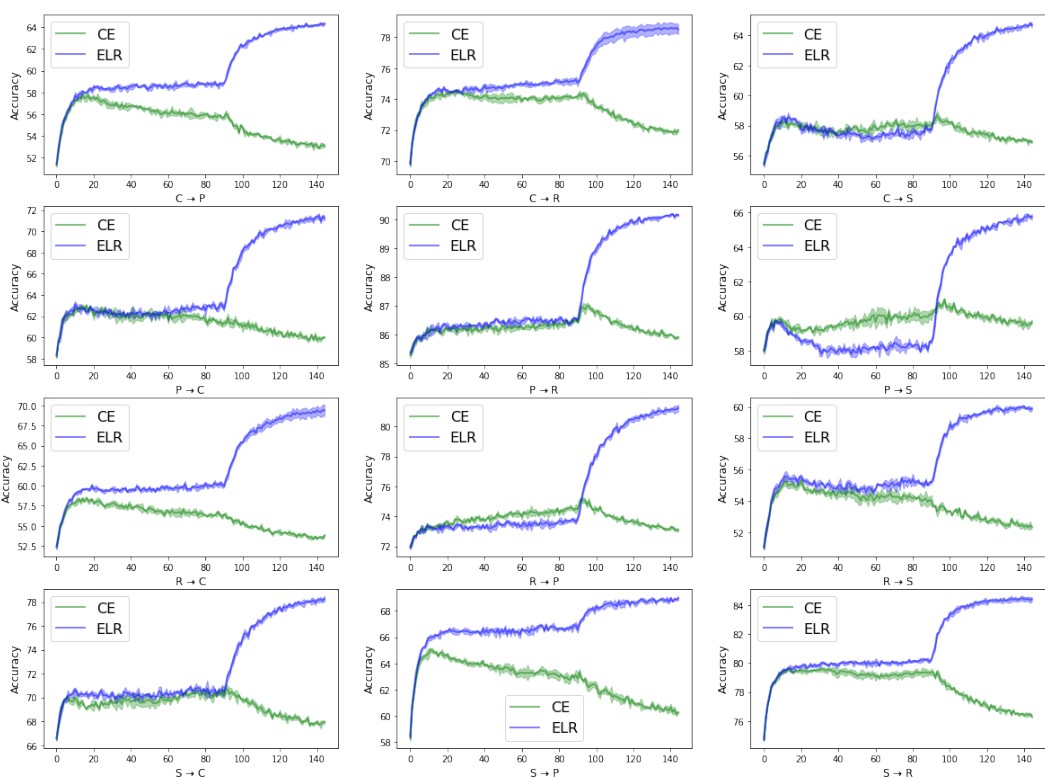

Figure 10: The source models are used to initialize the classifiers and annotate unlabeled target data. As the classifiers memorize the unbounded label noise very fast, we evaluate the prediction accuracy on target data every batch for the first 90 steps. After the 90 steps, we evaluate the prediction accuracy for every 0.3 epoch. We use the CE and ELR to train the classifiers on the labeled target data, shown in solid green lines and solid blue lines, respectively.

## G  EXPERIMENTAL DETAILS

In this section, we additionally show the overall training process of our method, illustrated in Figure 11 and in Algorithm 1. Besides, we provide more experimental information of our paper in details.

**Datasets.** We use four benchmark datasets, which have been widely utilized in the Unsupervised Domain Adaptation (UDA) (Long et al., 2015; Tan et al., 2020; Wang et al., 2022) and Source-Free

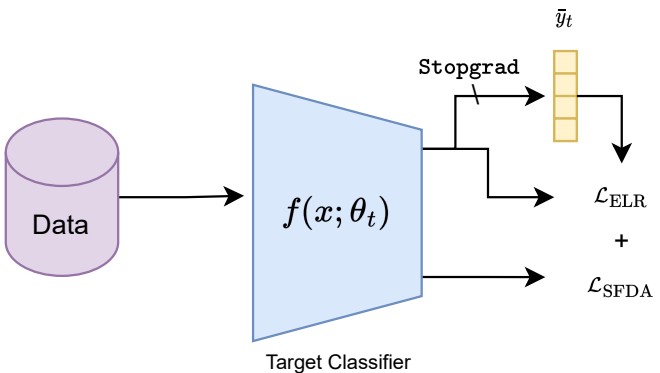

Figure 11: Overview of the SFDA problem and our method.

---

**Algorithm 1: SFDA ELR** - Source Free Domain Adaptation with ELR

---

**Input:** Source Pre-Trained Model: $f(x; \theta_0)$, Target Data: $\mathcal{X}_t(x_t)$, Training Epochs: T

1 Initialize a prediction bank $\mathcal{Y}$ with $\bar{y}_0 = \mathbf{0}$
2 **for** *epoch=1* **to** T **do**
3     **for** *iterations t=1,2,3,...* **do**
4         Compute the SFDA objective $\mathcal{L}_{\text{SFDA}}$ (depends on concrete SFDA algorithms)
5         Update the prediction bank $\mathcal{Y}$: $\bar{y}_t = \beta \bar{y}_{t-1} + (1 - \beta) f(x_t; \theta_t)$
6         Compute the ELR regularization $\mathcal{L}_{\text{ELR}}$: $\mathcal{L}_{\text{ELR}} = \log(1 - \bar{y}_t^\top f(x_t))$
7         Compute the total loss: $\mathcal{L} = \mathcal{L}_{\text{SFDA}} + \lambda \mathcal{L}_{\text{ELR}}$
8         Update the parameters of $f(\theta_t)$ via $\mathcal{L}$
9     **end**
10 **end**

**Output:** Target Adapted Model $f(x; \theta_T)$

---

Table 4: Accuracies (%) on Office-31 for ResNet50-based methods.

| Method | SF | A→D | A→W | D→W | W→D | D→A | W→A | Avg |
|---|---|---|---|---|---|---|---|---|
| MCD (Saito et al., 2018b) | ✗ | 92.2 | 88.6 | 98.5 | **100.0** | 69.5 | 69.7 | 86.5 |
| CDAN (Long et al., 2018) | ✗ | 92.9 | 94.1 | 98.6 | **100.0** | 71.0 | 69.3 | 87.7 |
| MDD (Zhang et al., 2019b) | ✗ | 90.4 | 90.4 | 98.7 | 99.9 | 75.0 | 73.7 | 88.0 |
| BNM (Cui et al., 2020) | ✗ | 90.3 | 91.5 | 98.5 | **100.0** | 70.9 | 71.6 | 87.1 |
| DMRL (Wu et al., 2020) | ✗ | 93.4 | 90.8 | 99.0 | **100.0** | 73.0 | 71.2 | 87.9 |
| BDG (Yang et al., 2020) | ✗ | 93.6 | 93.6 | 99.0 | **100.0** | 73.2 | 72.0 | 88.5 |
| MCC (Jin et al., 2020) | ✗ | 95.6 | 95.4 | 98.6 | **100.0** | 72.6 | 73.9 | 89.4 |
| SRDC (Tang et al., 2020) | ✗ | 95.8 | 95.7 | 99.2 | **100.0** | 76.7 | 77.1 | 90.8 |
| RWOT (Xu et al., 2020) | ✗ | 94.5 | 95.1 | **99.5** | **100.0** | **77.5** | 77.9 | 90.8 |
| RSDA-MSTN (Gu et al., 2020) | ✗ | **95.8** | **96.1** | 99.3 | **100.0** | 77.4 | **78.9** | **91.1** |
| Source Only | ✓ | 80.8 | 76.9 | 95.3 | 98.7 | 60.3 | 63.6 | 79.3 |
| **+ELR** | ✓ | 90.9 | 89.0 | 98.2 | **100.0** | 67.1 | 64.1 | 84.9 |
| SHOT (Liang et al., 2020) | ✓ | 94.0 | 90.1 | 98.4 | 99.9 | 74.7 | 74.3 | 88.6 |
| **+ELR** | ✓ | 94.9 | 91.6 | 98.7 | **100.0** | 75.2 | 74.5 | 89.3 |
| G-SFDA (Yang et al., 2021b) | ✓ | 85.9 | 87.3 | 98.6 | 99.8 | 71.4 | 72.1 | 85.8 |
| **+ELR** | ✓ | 86.9 | 87.8 | 98.7 | 99.8 | 71.4 | 72.9 | 86.2 |
| NRC (Yang et al., 2021a) | ✓ | 93.7 | 93.8 | 97.8 | **100.0** | 75.5 | 75.6 | 89.4 |
| **+ELR** | ✓ | 93.8 | 93.3 | 98.0 | **100.0** | 76.2 | 76.9 | 89.6 |

Table 5: Optimal Hypermaraters ($\beta/\lambda$) on various datasets.

| Hyperparameters: $\beta/\lambda$ | Office-31 | Office-Home | VisDA | DomainNet |
|---|---|---|---|---|
| ELR only | 0.9/− | 0.99/− | 0.99/− | 0.9/− |
| ELR + SHOT | 0.7/1.0 | 0.6/3.0 | 0.6/25 | 0.7/7.0 |
| ELR + G-SFDA | 0.8/1.0 | 0.9/1.0 | 0.5/7.0 | 0.8/12.0 |
| ELR + NRC | 0.5/1.0 | 0.6/3.0 | 0.5/3.0 | 0.8/3.0 |

Domain Adaptation (SFDA) (Liang et al., 2020) scenarios, to verify the effectiveness of leveraging the early-time training phenomenon to address unbounded label noise. Office-31 (Saenko et al., 2010) contains $4,652$ images in three domains (Amazon, DSLR, and Webcam), and each domain consists of 31 classes. Office-Home (Venkateswara et al., 2017) contains $15,550$ images in four domains (Real, Clipart, Art, and Product), and each domain consists of 65 classes. VisDA (Peng et al., 2017) contains 152K synthetic images and 55K real object images with 12 classes. DomainNet (Peng et al., 2019) contains around 600K images in six different domains (Clipart, Infograph, Painting, Quickdraw, Real and Sketch). Following previous work Tan et al. (2020); Liu et al. (2021a), we select 40 the most commonly-seen classes from four domains: Real, Clipart, Painting, and Sketch.

**Implementation.** We use ResNet-50 (He et al., 2016) for Office-31, Office-Home and DomainNet, and ResNet-101 (He et al., 2016) for VisDA as backbones. We adopt a fully connected (FC) layer as the feature extractor on the backbone and another FC layer as the classifier head. The batch normalization layer is put between the two FC layers and the weight normalization layer is implemented on the last FC layer. We set the learning rate to 1e-4 for all layers except for the last two FC layers, where we apply 1e-3 for the learning rate for all datasets. The training for source models are set to be consistent with the SHOT (Liang et al., 2020). The hyperparameters for ELR with self-training, ELR with SHOT, ELR with G-SFDA, and ELR with NRC on four different datasets are shown in Table 5. We note that for ELR with self-training, there is only one hyperparameter $\beta$ to tune. The hyperparameters for existing SFDA algorithms are set to be consistent with their reported values for Office-31, Office-Home, and VisDA datasets. As these SFDA algorithms have not reported their performance for DomainNet dataset, We follow the hyperparameter search strategy from their work (Liang et al., 2020; Yang et al., 2021a;b), and choose the optimal hyperparameters $\beta = 0.3$ for SHOT, $K = 5$ and $M = 5$ for NRC, and $k = 5$ for G-SFDA.

# H    Memorization Speed Between Label Noise in SFDA and in Conventional LLN settings

Although ETP exists in both SFDA and conventional LLN scenarios, the memorization speed for them is still different. Specifically, the target classifiers memorize noisy labels much faster in the SFDA scenario. It has already been shown that it takes many epochs before classifiers start memorizing noisy labels in conventional LLN scenario (Liu et al., 2020; Xia et al., 2020a). We highlight that the main factor causing the difference is the label noise. To show it, we replace the unbounded label noise in SFDA with bounded random label noise, and we keep the other settings unchanged as introduced in 4. To replace the unbounded label noise with bounded random label noise, we use the source model to identify mislabeled target samples, then we assign random labels to these mislabeled samples. Figure 13 and Figure 14 show the learning curves on Office-Home and Office-31 datasets with unbounded label noise and random bounded label noise. To better visualize the learning curves with unbounded label noise, we re-plot Figures 13-14 with different y scale in Figures 15-16. These figures demonstrate that target classifiers memorizing noisy labels with unbounded label noise is much faster than noisy labels with random bounded label noise. The classifiers with bounded label noise (colored in red) are expected to memorize all noisy labels eventually. As illustrated in Figures 15-16, the classifiers with unbounded label noise (colored in green) show that the noisy labels are already memorized. We note that for the first 90 steps, the prediction accuracy is evaluated every batch, while the prediction accuracy is evaluated every 0.3 epoch after that time. Therefore, for unbounded label noise, target classifiers start memorizing the noisy labels within the first epoch (consisting of more than 90 batches).

There are some existing LLN methods such as PCL (Zhang et al., 2021) to purify noisy labels every epoch based on ETP. Due to this difference, these LLN methods are not helpful to solving label noise in SFDA as they are not able to capture the benefits of ETP. Our empirical results in Section 5.1 can support this argument. We also note that PCL does not suffer from the fast memorization speed and is able to capture the benefits of ETP in conventional LLN settings. As we indicated in Figures 13-14, it takes much longer time (more than a few epochs) for target classifiers to start memorizing bounded noisy labels. We hope these insights can motivate the researcher to consider memorization speed and design algorithms better for SFDA.

# I    Additional Analysis of ELR and a standard SFDA method - NRC

In this section, we will theoretically and empirically compare ELR and NRC in detail. Specifically, NRC (Yang et al., 2021a) is a well-known SFDA method that explores the neighbors of target data by graph-based methods and utilizes these neighbors' information to correct the target data's pseudo-label, in order to boost the SFDA performance. The proposed NRC loss has the following form:

$$\ell_{\mathbf{NRC}} = \mathcal{L}_{div} + \mathcal{L}_{\mathcal{N}} + \mathcal{L}_E + \mathcal{L}_{self}$$

with:

$$\mathcal{L}_{div} = \sum_{k=1}^{K} KL(\bar{p_k} || q_k)$$

the diversity loss where $\bar{p_k}$ is the empirical label distribution and $q$ is a uniform distribution; and

$$\mathcal{L}_{\mathcal{N}} = -\frac{1}{n_t} \sum_i \sum_{m \in \mathcal{N}_M^i} A_{im} \mathcal{S}_m^{\mathsf{T}} h(\mathbf{x}_i)$$

the neighbors loss, where $m$ is the index of the $m$-th nearest neighbors of $\mathbf{x}_i$, $\mathcal{S}_m$ is the $m$-th item in memory bank $\mathcal{S}$, $A_{im}$ is the affinity value of $m$-th nearest neighbors of input $\mathbf{x}_i$ in the feature space.

$$\mathcal{L}_E = -\frac{1}{n_t} \sum_i \sum_{m \in \mathcal{N}_M^i} \sum_{j \in E_N^m} r \mathcal{S}_n^{\mathsf{T}} h(\mathbf{x}_i)$$

the expanded neighbors loss, where $E_N^m$ contain the $N$-nearest neighbors of neighbor $m$ in $\mathcal{N}_M$.

$$\mathcal{L}_{self} = -\frac{1}{n_t} \sum_i^{n_t} \mathcal{S}_i^{\mathsf{T}} h(\mathbf{x}_i)$$

the self-regularization loss, where $\mathcal{S}_i$ means the stored prediction in the memory bank, a constant vector and is **identical** to the $h(\mathbf{x}_i)$ as in NRC they update the memory banks **before** the training.

## I.1 THEORETICAL ANALYSIS OF NRC'S SELF-REGULARIZATION TERM COMPARED TO ELR

To emphasize the novelty of our proposed ELR in SFDA problems, we will compare the original formulas and also the gradients of ELR and NRC's self-regularization (SR) term in detail. And then, we will explain why NRC can not benefit from the ETP only by adopting the SR term. As we formulate in the main paper, we can represent the ELR loss and the SR loss as follows:

$$\mathcal{L}_{\text{ELR}}(\theta_t) = \log(1 - \bar{y}_t^\top f(\mathbf{x}; \theta_t)) \tag{38}$$

and

$$\mathcal{L}_{\text{SR}}(\theta_t) = -\hat{y}_t^\top f(\mathbf{x}; \theta_t) \tag{39}$$

where $\bar{y}_t = \beta \bar{y}_{t-1} + (1-\beta) f(\mathbf{x}; \theta_t)$ in ELR is the moving average prediction for $\mathbf{x}$, and $\hat{y}_t = f(\mathbf{x}; \theta_t)$ in SR is the constant vector copied from the **current** training step's prediction. Besides, the gradients of ELR and SR are:

$$\frac{\mathrm{d}\mathcal{L}_{\text{ELR}}(\theta_t)}{\mathrm{d}f(\mathbf{x}; \theta_t)} = -\frac{\bar{y}_t}{1 - \bar{y}_t^\top f(\mathbf{x}; \theta_t)} \tag{40}$$

and

$$\frac{\mathrm{d}\mathcal{L}_{\text{SR}}(\theta_t)}{\mathrm{d}f(\mathbf{x}; \theta_t)} = -\hat{y}_t \tag{41}$$

The motivation of the SR term is to emphasize the ego feature of current prediction and, therefore, to reduce the potential impact of noisy neighbors, whereas the ELR proposed in this paper considers the changes of prediction quality during the training process and aims to encourage the model prediction to stick to the early-time predictions for each data point.

As shown in Eq. ( 38) and Eq. ( 39), we can directly observe that ELR involves the previous training step's prediction information in loss (included in $\bar{y}_t$), however, SR leverages only the prediction result of current step.

Besides, if we further look at the gradient formulas of these two losses and analyze the back-propagation process, we can find that the gradient $\frac{\mathrm{d}\mathcal{L}_{\text{ELR}}(\theta_t)}{\mathrm{d}f(\mathbf{x};\theta_t)}$ increases as the model prediction closes to the target $\bar{y}_t$, which will further force the prediction $f(\mathbf{x}; \theta_t)$ close to $\bar{y}$ thanks to the large magnitude of the gradient. And this will help with the utilization of early-time predictions and ETP. However, the gradient of $\mathcal{L}_{\text{SR}}$ is a constant vector with values of prediction logits, which could be very small. So when $\frac{\mathrm{d}\mathcal{L}_{\text{SR}}(\theta_t)}{\mathrm{d}f(\mathbf{x};\theta_t)}$ is small, SR term can be easily overwhelmed by the other loss terms that favour fitting incorrect pseudo labels, leading to poor performance.

The above analysis shows a fundamentally different difference between SR and ELR. Specifically, SR does not utilize ETP and cannot handle the unbounded label noise either.

## I.2 EMPIRICAL ANALYSIS OF ELR AND NRC IN TERMS OF THE UTILIZATION OF ETP

In addition to the above theoretical analysis for the loss functions, we also observed the same conclusion through experiments. As shown in Figure 12, we observe that thanks to the update of the pseudo-label with the process of adaptation in the SFDA method, overall, NRC can obtain a model with relatively high accuracy on the target domain. However, the performance drop still exists when using the NRC method alone, which can be effectively avoided by adding the ELR term. This confirms that ELR can effectively leverage ETP and avoid the problem of noisy label memorization.

## J ADDITIONAL DISCUSSION OF PSEUDO-LABEL PURIFICATIONS IN SFDA AND LLN APPROACHES

In this section, we will further discuss the similarities and the differences between the LLN approaches and the pseudo-label purification processes proposed in current SFDA methods.

The main similarity between the existing SFDA approaches and the LLN methods is that both research fields have to deal with data with noise, aiming to get a model with promising performance. As

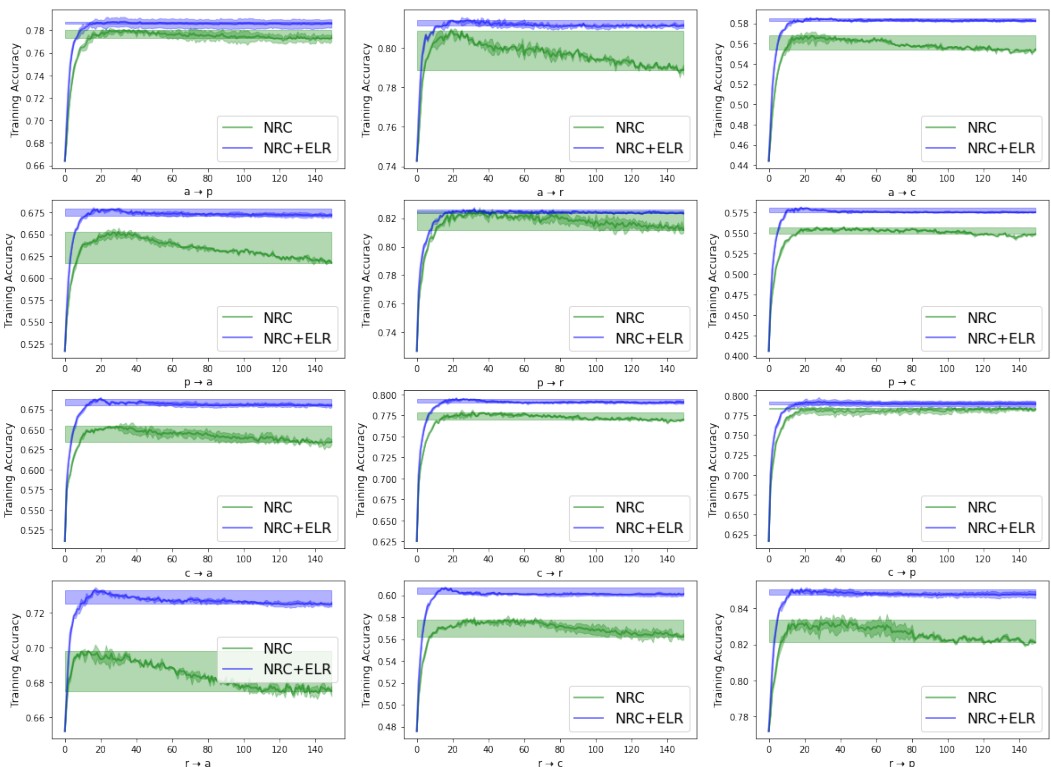

Figure 12: Fine-Grained Training Accuracy of NRC and NRC + ELR on Office-Home dataset. The solid green lines represent the training process of NRC, whereas the solid blue lines represent the training process of NRC with ELR term. The colored bands represent the performance drop.

for the differences, they can be mainly divided into the following aspects. From the perspective of motivations, most of the existing SFDA approaches are developed under the domain adaptation setting. They study how to best exploit the distribution relationship between the source and target domains in the absence of source data, so as to achieve domain adaptation better. Their motivation is to investigate how to better assign the pseudo-label. In contrast, LLN is an independent field that mainly studies, given a set of noisy data, how to deal with the label noise, conduct the model training, and obtain a noise-robust model with better performance. Traditionally, the study of LLN does not involve assumptions about the data domain or source model. Meanwhile, there are more in-depth and rigorous studies (theoretical and methodological) on the types of noises, and how to handle and exploit them. From the perspective of the methodology, in order to obtain a higher quality pseudo-label, many SFDA methods heuristically use clustering or neighbor features to correct the pseudo-labels, and use the corrected labels to perform a normal supervised learning. The current SFDA methods focus on the explicit pseudo-label purification process, which can be summarized as noisy label correction. However, for LLN, the noisy label correction is just a research sub-branch. LLN also includes many other research directions, such as studies of different label noise types, research about how to utilize and even benefit from label noise in the training process, and how to train the model more robustly. Many noise-robust loss functions and related theoretical analyses have been developed.

We would like to emphasize that the motivation of our paper is to investigate how to study SFDA from the perspective of learning with label noise. We combine the characteristics of SFDA with the LLN approaches and discover the unbounded nature of label noise in SFDA. Further, we rigorously distinguish which LLN methods can help SFDA problems and which approaches are limited in their use in SFDA. We believe that the studies of LLN can open new avenues for the research of SFDA and bring more ideas and inspiration to the design of the SFDA algorithm.

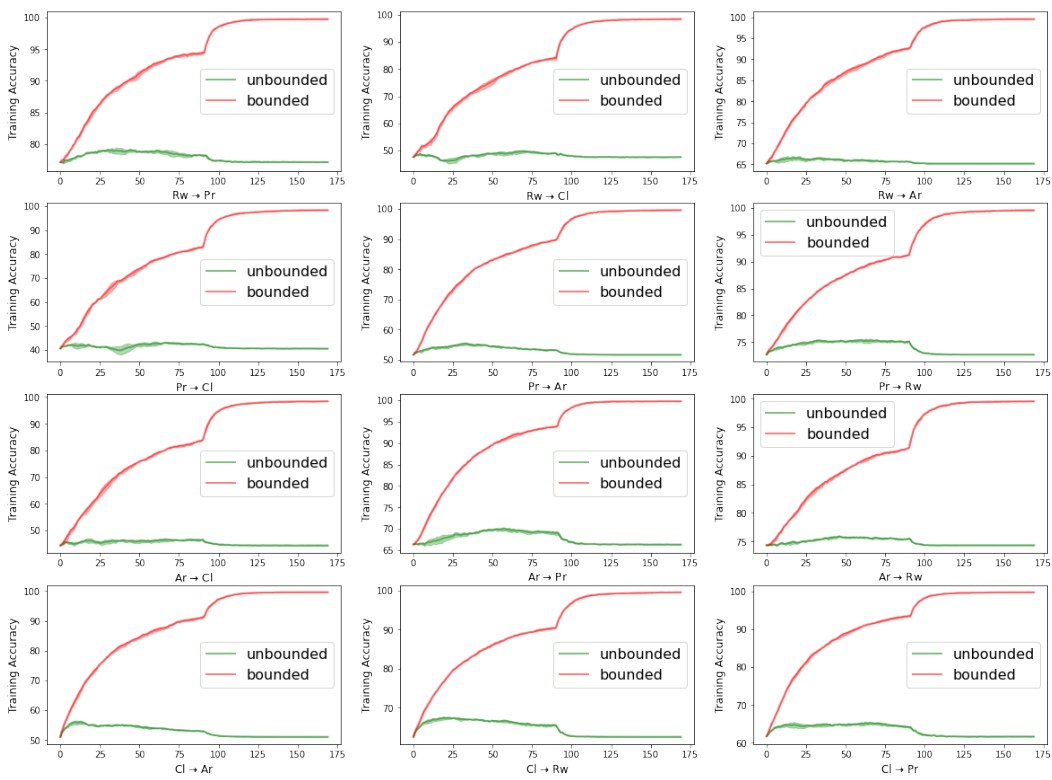

Figure 13: Training accuracy on Office-Home dataset. The solid green lines represent the unbounded label noise in SFDA, whereas the solid red lines represent the bounded label noise.

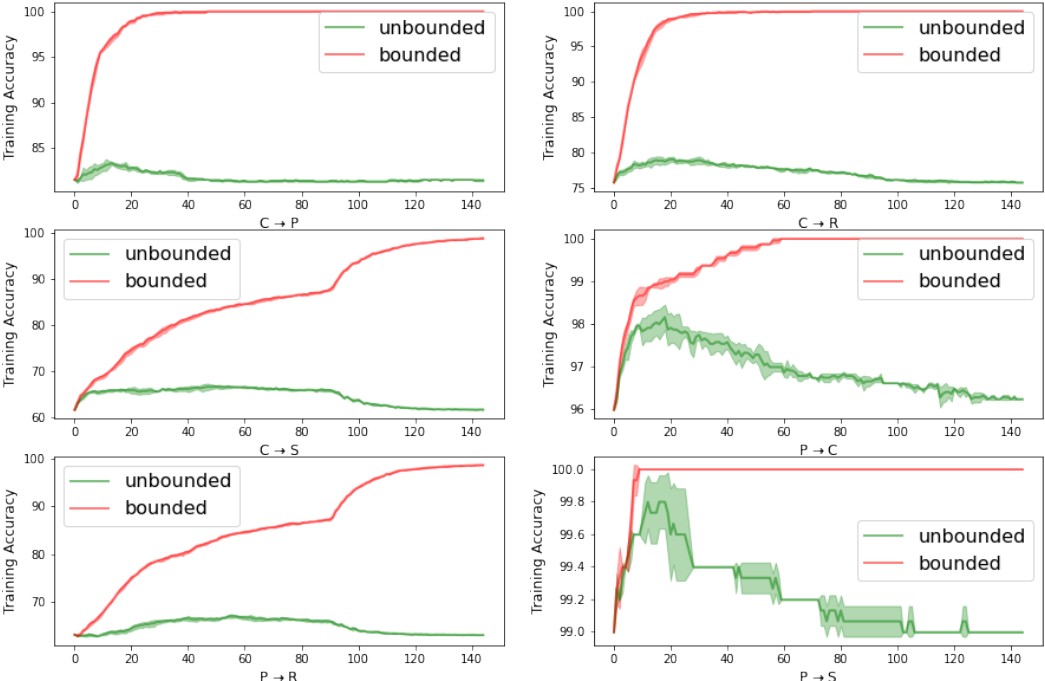

Figure 14: Training accuracy on Office-31 dataset. The solid green lines represent the unbounded label noise in SFDA, whereas the solid red lines represent the bounded label noise.

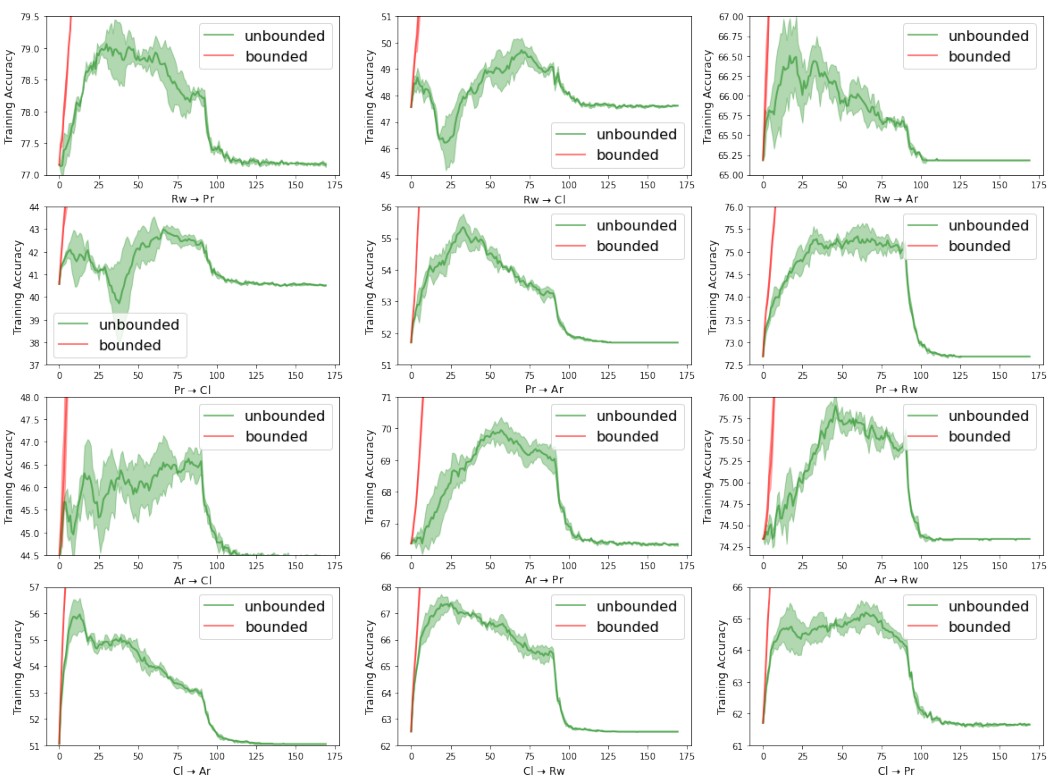

Figure 15: Figure 13 with different y-scale to better show learning details of the unbounded label noise.

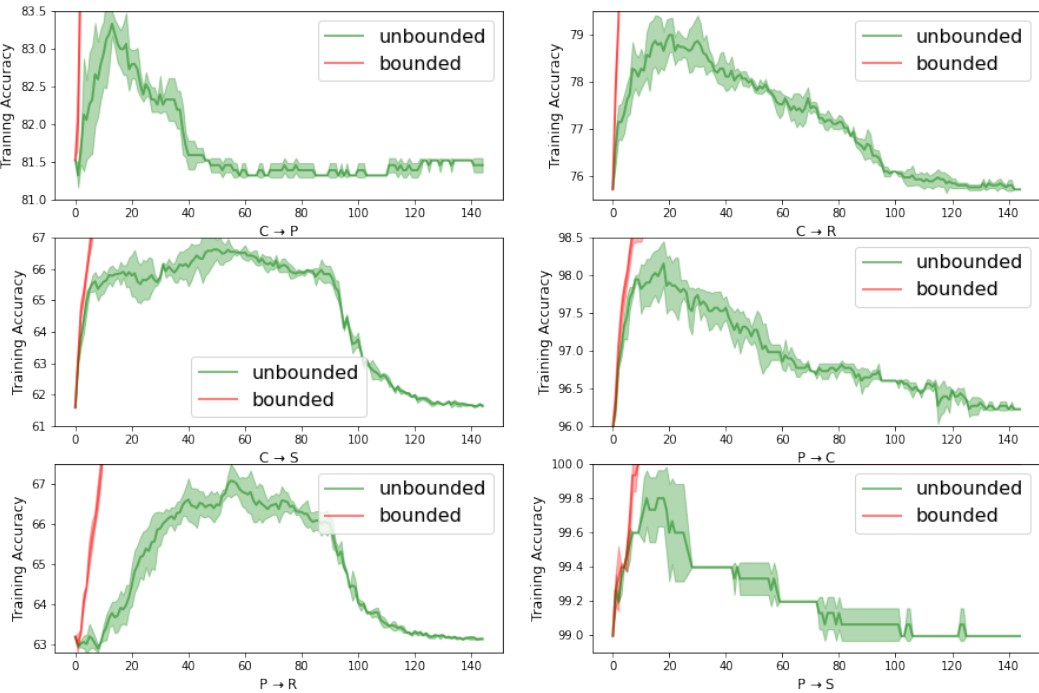

Figure 16: Figure 14 with different y-scale to better show learning details of the unbounded label noise.

