# OpenReview forum: "When Source-Free Domain Adaptation Meets Learning with Noisy Labels"
_ICLR.cc/2023/Conference — ICLR 2023 notable top 25%_

### Official Review · Reviewer_nSg6 · 2022-10-16

**Confidence:** 3
**Correctness:** 3
**Technical Novelty And Significance:** 3
**Empirical Novelty And Significance:** 3
**Recommendation:** 6

**Clarity, Quality, Novelty And Reproducibility:**

Overall, the paper is well-written and easy to follow. Below, I have a few questions regarding unclear points that I encountered.

- Regarding Eq. (1), having a plot of the graph of the mislabeling rate as a function of $\\Delta$ may help readers to understand that the mislabeling rate increases as the magnitude of domain shift increases. It should be fine to have the plot in the supplementary material.
- In Theorem 3.1, it is not clear why we can assume $\\Delta$ is positively correlated with the vector $\\mu_2 - \\mu_1$ without loss of generality. (In any cases, I think we may "lose generality" because the aim of this theorem is to show an example where the mislabeling rate is unbounded.)
- Lemma 3.2 and the following paragraph do not explain $\\ell_\\mathrm{LLN}$ in its full detail so that the claim of the lemma is not clear enough to the readers not familiar with the existing work on LLN. It looks essential to introduce the background concept that a line of researches on LLN uses noise-robust loss functions instead of the usual classification loss to learn a noise robust classifier.
- In Figure 2, why do both green and blue curves suddenly change the tendency at epoch 90? It is explained "After 90 steps, we evaluate the prediction accuracy for every 0.33 epoch," but I do not see this affect that much.
- In Figure 2, I would like to see if well-known noise-robust loss functions such as GCE (Zhang & Sabuncu, 2018) and so on suffer from the memorizing issue as well. Adding a few more loss functions to Figure 2 would be comprehensive to argue the importance of considering the unbounded noise rate.
- In the experimental results, I do not see why the performance of UDA methods are uniformly worse than source-free domain adaptation (e.g. in Table 1). My intuition tells that the problem setting of UDA provides a learner more information about source data and hence she could leverage the richer information to improve the performance over source-free domain adaptation. Perhaps it gets off the topic here, but I appreciate if the authors can comment on this issue.

**Strength And Weaknesses:**

### Strengths

- **Nice theory to demonstrate the issue of unbounded noises.** Although the setup is limited to a simple Gaussian case, the authors provide a nice theoretical illustration of how the labeling error of the pseudo-labeling in source-free domain adaptation can be arbitrarily large.
- **Simple remedy to existing source-free domain adaptation.** The proposed remedy to the unbounded noise is to use the moving average of the label prediction over training epochs, which is practically very simple to use. So far, this method has been used in the context of learning from label noises, but its applicability and effectiveness in source-free domain adaptation has not been witnessed. The authors contribute in this line to show ETP can be observed in the unbounded noise case as well.

### Weaknesses

Basically, I do not see any weaknesses of this paper except for small unclear points. I point out them in the subsequent "Clarity" section.

**Summary Of The Paper:**

This paper considers source-free domain adaptation, where a learner has access to only a source model and unlabeled target data due to privacy concerns regarding source data. Existing practices in source-free domain adaptation is to use the source model to pseudo-label target data, which may cause mislabeling occasionally, and moreover, the mislabeling probability is much higher than the noise probability that is usually assumed in learning from label noises. Indeed, the authors show a theoretical statement that the mislabeling probability can be arbitrarily large and approaching 1 under a simple setting. Then, the authors reveal that ETP (early-time training phenomenon), which observes that a learner with fewer training epochs can avoid overfitting and memorizing the mislabeled data perfectly, can be observed in the unbounded noise case as well. By leveraging this fact, the authors propose to use early learning regularization (ELR), which uses the moving average of the label prediction over training epochs to mitigate the aforementioned overfitting issue, with existing source-free domain adaptation methods. As a result, the existing methods can be improved with ELR.

**Summary Of The Review:**

The paper contributes to source-free domain adaptation from a new viewpoint, learning with label noises, which should be acknowledged. The overall quality of the manuscript is above the border. A few points I mentioned in "Clarity" section can be addressed, which I believe makes the paper much better.

---

> ### Author Response · Authors · 2022-11-14
> **Response to Reviewer nSg6 (2/2)**
>
> >**Q4**: **In Figure 2, why do both green and blue curves suddenly change the tendency at epoch 90? It is explained "After 90 steps, we evaluate the prediction accuracy for every 0.33 epoch," but I do not see this affect that much.**
>
> Thanks for your question. We would like to clarify that **the unit of the horizontal coordinate is step instead of epoch**. When the horizontal coordinate (step) is less than 90 and more than 90, the length of the training process (evaluation interval size) represented by each step is different. For example, suppose **a training epoch contains 900 batches**. Then, when the horizontal coordinate step is less than 90, one step represents an evaluation right after the update process of **every single training batch**, whereas when it is greater than 90, each step represents the evaluation result **every 0.33 training epoch, that is, an evaluation per 300 training batches**. At the beginning of the adaptation training, the model's performance changes very quickly, which is why there is a sudden change in the curve at **step=90**.
>
> We agree that the caption of Figure 2 is a little confusing, and we clarified this point in the updated manuscript.
>
> >**Q5**: **In Figure 2, I would like to see if well-known noise-robust loss functions such as GCE (Zhang & Sabuncu, 2018) and so on suffer from the memorizing issue as well. Adding a few more loss functions to Figure 2 would be comprehensive to argue the importance of considering the unbounded noise rate.**
>
> Thanks for your insightful remark and suggestions. **We conducted additional experiments with GCE loss and observed a similar performance drop during the training process, shown in the updated Figure 2.** This observation indicates that the conventional noise-robust losses will also suffer from the memorizing issue and can not benefit from ETP, which further emphasizes the importance of introducing ELR in SFDA. We agree that the existence of unbounded noise is a possible cause of the performance drop under GCE loss, which therefore confirms the necessity of considering the unbounded noise rate in SFDA. And this is consistent with our experimental results.
>
> The GCE loss curve and the corresponding explanations have been updated in the manuscript.
>
> >**Q6**: **In the experimental results, I do not see why the performance of UDA methods are uniformly worse than source-free domain adaptation (e.g. in Table 1). My intuition tells that the problem setting of UDA provides a learner more information about source data and hence she could leverage the richer information to improvethe performance over source-free domain adaptation. Perhaps it gets off the topic here, but I appreciate if the authors can comment on this issue.**
>
> Thanks for your remark. We would like to explain that the results of UDA methods in Table 1-4 come from previous SFDA and UDA papers [1-2]. We agree that the problem setting of UDA, intuitively, can benefit from more source data information and hence could leverage the richer information to improve the performance than SFDA. However, the current state of research is, because of the lack of source data, SFDA methods focus on how to maximize the use of information from the target input data and source model. They go beyond the traditional UDA research and combine the techniques from other domains, such as pseudo-label, clustering, and self-supervised learning, to essentially improve the performance of models on the target domain. The methods used in these SFDAs are not included in the UDA methods shown in the table. In fact, this is a very insightful, interesting observation and is also an open topic. To the best of our knowledge, there is not much research exploring whether and how we can use SFDA methods to improve the performance of UDA in the reverse direction.
>
>
> **References:**
>
> [1] *Shiqi Yang, Yaxing Wang, Joost van de Weijer, Luis Herranz, Shangling Jui: Exploiting the Intrinsic Neighborhood Structure for Source-free Domain Adaptation. In NeurIPS 2021*
>
> [2] *Hong Liu, Jianmin Wang, Mingsheng Long: Cycle Self-Training for Domain Adaptation. In NeurIPS 2021*

---

> ### Author Response · Authors · 2022-11-14
> **Response to Reviewer nSg6 (1/2)**
>
> We would like to sincerely appreciate the reviewer nSg6 for the insightful review and constructive suggestions. We will fully address the concerns about the unclear points below.
>
> >**Q1**: **Regarding Eq. (1), having a plot of the graph of the mislabeling rate as a function of $\Delta$ may help readers to understand that the mislabeling rate increases as the magnitude of domain shift increases. It should be fine to have the plot in the supplementary material.**
>
> Thanks for your suggestion. We agree that a graph can help understand the relationship between the degree of domain shift and the corresponding mislabeling rate. We added a plot (shown as **Figure 8**) and updated the corresponding explanation in the **Section B of Appendix**.
>
> We can observe in the plot that the labeling error increases with the magnitude of domain shift $\Delta$ increasing, which is consistent with our previous explanation that "the magnitude of the domain shift inherently controls the mislabeling error for target data".
>
> >**Q2**: **In Theorem 3.1, it is not clear why we can assume $\Delta$ is positively correlated with the vector $\boldsymbol{\mu_2}-\boldsymbol{\mu_1}$ without loss of generality. (In any cases, I think we may "lose generality" because the aim of this theorem is to show an example where the mislabeling rate is unbounded.)**
>
> Thanks for your remarks. We would like to explain that:
>
> - $\Delta$ represents the domain shift, and here we use "positively correlated" to indicate the scalar product of $\Delta$ and $\boldsymbol{\mu_2}-\boldsymbol{\mu_1}$ is positive, which only aims to simplify the computational process and the presentation of the results.
> - We claim "without loss of generality" because we can use a similar symmetric proof procedure in the negatively correlated situation (when $\Delta^{T}(\boldsymbol{\mu_2}-\boldsymbol{\mu_1}) \leq 0$) to obtain the same unbounded conclusion.
> - In fact, as long as the dot product $\Delta^{T}(\boldsymbol{\mu_2}-\boldsymbol{\mu_1})$ is not equal to 0, i.e., if the domain shift does not happen along the perfect decision boundary and will affect the Bayesian optimal classifier, we can always find a region R where the unbounded mislabeling rate phenomenon exists.
>
> To facilitate the reader's understanding, we added some explanation in Theorem 3.1 and Theorem B.1. We also completed the mislabeling rate plot for $\Delta$ negatively correlated and perpendicular with $\boldsymbol{\mu_2}-\boldsymbol{\mu_1}$ situations on the newly added Figure 8. We hope that such an illustration clarifies our assumption's meaning and helps to understand that these assumptions will not influence Theorem 3.1's conclusion.
>
> >**Q3**: **Lemma 3.2 and the following paragraph do not explain $l_{LLN}$ in its full detail so that the claim of the lemma is not clear enough to the readers not familiar with the existing work on LLN. It looks essential to introduce the background concept that a line of researches on LLN uses noise-robust loss functions instead of the usual classification loss to learn a noise robust classifier.**
>
> We appreciate your insightful remarks and suggestions. We would explain that in Appendix D of the original manuscript, we have given a brief description of the noise-robust loss based LLN methods and the concrete instances of $\ell_{LLN}$ involved in Lemma 3.2 before proving it.
>
> However, we agree that more explanation and background concepts of noise-robust loss based LLN methods will make it easier for people unfamiliar with this topic to understand lemma 3.2 and the whole paper. We, therefore, made the following changes:
>
> - We re-adjusted the introduction of the LLN methods in the related work.
> - We emphasized the meaning of $\ell_{LLN}$ while introducing Lemma 3.2 in the main paper.
> - Due to the limited space, we added a section in Appendix (D.1) to further present the related LLN methods in detail.
>
> We hope the updated manuscript is clear and can address your concerns.

---

> ### Author Response · Authors · 2022-11-18
> **Discussion**
>
> Dear Reviewer nSg6,
>
> We thank you again for your constructive suggestions, insightful comments, and valuable time.
>
> We have revised the manuscript and further explained the unclear points according to your suggestions. As the discussion deadline is approaching, we would appreciate it if you could let us know whether our responses have addressed your questions and concerns. We are happy to provide any further information or clarifications if needed.
>
> Best,
>
> Paper3266 Authors

---

> > ### Comment · Reviewer_nSg6 · 2022-11-25
> > **Reply**
> >
> > I thank the authors for addressing the comments very carefully.
> > I think some unclear points in the original manuscript have been modified thoroughly in the revision.
> >
> > By the way, I found on p.4 of the modified pdf that the notation $\\mathcal{M}\_{d \\times d}$ would be used without the definition. Can you add it?

---

> > > ### Author Response · Authors · 2022-11-25
> > > **Thanks for the feedback**
> > >
> > > Dear Reviewer nSg6,
> > >
> > > We are glad that our previous responses and the revised manuscript addressed your concerns, and we really appreciate your rigorous advice. We want to clarify that the $\mathcal{M}_{d\times d}(\mathbb{R})$ mentioned in the paper represents the set of all $d$-by-$d$ real matrices. Given that we cannot update the manuscript in the second discussion stage, we will add this explanation in the final version.
> > >
> > > Thank you again for your detailed feedback and the valuable time you have invested.
> > >
> > > Best,
> > >
> > > Paper3266 Authors

---

### Official Review · Reviewer_3S6g · 2022-10-21

**Confidence:** 3
**Correctness:** 3
**Technical Novelty And Significance:** 2
**Empirical Novelty And Significance:** 2
**Recommendation:** 6

**Clarity, Quality, Novelty And Reproducibility:**

Clarity: The paper was generally clear and easy to follow.
More detailed explanations of the experiment protocol would be helpful.

Quality: I think the paper holds good quality. It presents interesting motivations and demonstrates both theoretical and empirical justifications for the proposed method.

Novelty: The originality of the work seems a bit limited. While the idea of levering ETP for SFDA seems novel, they used the recently established ELR. In addition, it shares some similarities with the self-regularization term previously used in the NRC.

Reproducibility: The paper does not provide enough details to reproduce their experiment results.
I assume their attached codes may resolve the reproducibility issue.


**Strength And Weaknesses:**

Strengths
- I think the idea of formulating the SFDA problem as LLN is interesting. The authors also theoretically showed why the existing LLN methods cannot address the noise in SFDA due to the fundamental differences.
- The theoretical and empirical justification of ETP in the SFDA problem with the unbounded label noise seems to provide solid grounds for the proposed method.
- Leveraging ETP with ELR term seems simple but effective across a wide range of SFDA algorithms and benchmark datasets.

Weaknesses
- While the previous SFDA methods do not explicitly formulate the SFDA problem as LLN, I think they are still partially based on the LLN methods. As the authors explained in the related work section, the SFDA methods leverage the target “noisy” pseudo labels and also some type of pseudo-label purification processes as in the LLN methods. It would be great if the authors can provide more clear explanations of the similarities and differences between their motivations.
- The proposed early learning regularization (ELR) term encourages the model prediction to stick to the early-time predictions for each data point. Meanwhile, NRC contains a loss term, called self-regularization loss, which has a similar purpose of reducing the potential impact of noisy pseudo-labels and not ignoring the current prediction. Please provide comparative explanations between the two and how the ELR still improves the NRC (Table1).
- In Empirical Observations on Real-World Dataset, it seems the authors trained classifiers using the fixed noisy labels with cross-entropy loss. However, SFDA algorithms (e.g. NRC) gradually update pseudo-labels rather than using the fixed pseudo-labels. Please provide clarifications. It would be also nice if you can also show the training accuracy of SFDA algorithms in Figure 2.
- How did the authors get the baseline results? There seem to be some discrepancies compared to the results reported in their original papers. For example, NRC results seem lower than those reported in the original paper. The reported average results are 72.2 for Office-Home and 85.9 for VisDA-C. Then, the performance difference between NRC+ELR in Table1 (72.6) and the NRC in the original paper (72.2) seems marginal.
- Please provide a more detailed protocol for the experiments. For example, assuming that no labeled target data is available, how do you decide (1) when to stop the training and (2) hyperparameters for each dataset? If the authors hand-picked the hyperparameters for each dataset, it might have caused over-optimistic results.


**Summary Of The Paper:**

This paper tackles the problem of source-free domain adaptation (SFDA), where a pre-trained source model is adapted using unlabeled target domain data without accessing any source domain data. While previous works mainly focused on cluster assumption in the feature space, the authors propose a different perspective. They formulate the SFDA problem as learning with label noise (LLN). In addition, since label noise in SFDA is unbounded unlike the noise in conventional LLN scenarios, they leverage the early-time training phenomenon (ETP) with a simple early learning regularization (ELR) term. The authors demonstrate that it consistently improves existing SFDA algorithms by a large margin on four different SFDA benchmark datasets.

**Summary Of The Review:**

While the paper has its merits, unfortunately, it also has several issues which need to be addressed: (1) clarification of the novelty of the proposed method, (2) comparative explanations between the proposed method and self-regularization loss in NRC, and (3) detailed experiment protocols.

---

> ### Author Response · Authors · 2022-11-14
> **Response to Reviewer 3S6g (4/4)**
>
> > **Q5**: **Please provide a more detailed protocol for the experiments. For example, assuming that no labeled target data is available, how do you decide (1) when to stop the training and (2) hyperparameters for each dataset? If the authors hand-picked the hyperparameters for each dataset, it might have caused over-optimistic results.**
>
> Thanks for your comments. We would like to clarify that to ensure the fairness and generalizability of the experimental results, we followed the previous methods for the hyperparameters setting and selection[1-3], and that we did not conduct any hand-picking or heavy tuning process for the hyperparameters.
>
> More concretely, **for the first question about "when to stop the training"**, we just followed the baselines' (i.e., SHOT, G-SFDA, and NRC) training pipeline. We utilized the same training parameters, such as epoch number, batch size, and learning rate, provided in their code repositories and papers for both the baseline method and baseline + ELR. In fact, different from other methods that also leverage better model performance during the early training stages, like early stopping, ELR can constantly utilize the training information by adaptively updating pseudo labels to guide the model training. Therefore, we do not need to spend additional effort to decide when to stop training. More details of the experiment implementation can be found in Appendix G.
>
> **As for the concerns about the hyperparameter selection**, we would like to claim that we just follow the same hyperparameter selection scheme utilized in the previous SFDA methods [1-3]. For the experiments on Office-Home, Office-31 and VisDA, we utilized the best hyperparameters provided in the original papers for baseline algorithms [1-3]. As for the experiments on DomainNet, we first leveraged the tuning methods described in the original baseline papers to obtain their best hyperparameters, respectively. We then introduced the ELR term on top of that. Moreover, for the two hyperparameters related to ELR, we followed the hyperparameter analysis approach in [1]. As shown in Figure 3 from the main paper, we can observe that the ELR model's performance is not sensitive to different values of hyperparameters, and there is no need to tune them heavily. Besides, we can also note that our method outperforms the baseline in a wide range of hyperparameters, which further verifies that our experimental results obtained are generalizable and guaranteed.
>
> **References:**
>
> [1] *Shiqi Yang, Yaxing Wang, Joost van de Weijer, Luis Herranz, Shangling Jui: Exploiting the Intrinsic Neighborhood Structure for Source-free Domain Adaptation. In NeurIPS 2021*
>
> [2] *Shiqi Yang, Yaxing Wang, Joost van de Weijer, Luis Herranz, Shangling Jui: Generalized Source-free Domain Adaptation. In ICCV 2021*
>
> [3] *Jian Liang, Dapeng Hu, Jiashi Feng: Do We Really Need to Access the Source Data? Source Hypothesis Transfer for Unsupervised Domain Adaptation. In ICML 2020*
>
> [4] *Sheng Liu, Jonathan Niles-Weed, Narges Razavian, Carlos Fernandez-Granda: Early-Learning Regularization Prevents Memorization of Noisy Labels. In NeurIPS 2020*

---

> > ### Comment · Reviewer_3S6g · 2022-11-14
> > **Thank you for the detailed response**
> >
> > Thanks for the detailed response and for providing additional experiment results. Most of my concerns are addressed. Upon further consideration, I've decided to raise my score from 5 to 6. Generally, I think it's a solid paper with interesting motivations, theoretical analyses, and empirical results.

---

> > > ### Author Response · Authors · 2022-11-14
> > > **Thanks for your feedback**
> > >
> > > We sincerely thank you for the positive comments and also for recognizing the contributions of our work! Please let us know if you need any further information or clarifications.

---

> ### Author Response · Authors · 2022-11-14
> **Response to Reviewer 3S6g (3/4)**
>
> > **Q4**: **How did the authors get the baseline results? There seem to be some discrepancies compared to the results reported in their original papers. For example, NRC results seem lower than those reported in the original paper. The reported average results are 72.2 for Office-Home and 85.9 for VisDA-C. Then, the performance difference between NRC+ELR in Table1 (72.6) and the NRC in the original paper (72.2) seems marginal.**
>
> Thanks for your questions. We would like to explain this question and our experiment process in detail.
>
>  - In our paper, to fairly compare the different performances of the current SFDA method with and without the ELR term, we implemented the baseline methods by ourselves instead of directly copying numbers from the original papers. During the experiments of each baseline method, we utilized the same training parameters for baseline and baseline+ELR, to **make sure the only difference between our algorithm and the baseline is the ELR term**.
>  - More concretely, we utilized the GitHub repositories officially provided by the baseline methods. In the training process, we followed exactly the training procedure described in the original papers in order to reproduce the results as much as possible, using the provided best hyperparams and the officially given well-trained source models. In particular, we conduct experiments on DomainNet, a challenging dataset that few SFDA methods have touched. During the experiment, we first used the tuning methods described in the original papers to obtain the baselines' best hyperparameters. We then introduced the ELR term on top of that. We can observe that the proposed NRC+ELR method also improves significantly over NRC on DomainNet (3%).
>  - Thanks to your remark on the performance difference in Table1. We would like to clarify that, when we implemented the baseline method NRC using their official code, we found that there is a **neighborhood number smoothing trick** (line 223-226 in NRC_SFDA/office-home/train_tar.py from NRC official Github Repository), which **has not** been mentioned in their paper and has been leveraged **only on the Office-Home dataset**. More concretely, they enlarged the considered neighborhood number from 3 to 5 after half of the training process. To make the experimental procedure more consistent and simple, we did not adopt this trick (for both NRC and NRC+ELR) in our implementations and got the results shown in Table1 in the main paper. However, we can totally understand your concerns, and we, therefore, re-conducted the experiments of NRC and NRC+ELR with this neighborhood number smoothing trick on the Office-Home dataset. The results are shown in the table below. We can still observe a consistent improvement by the proposed method.
>
>     |  Method | Ar→Cl | Ar→Pr | Ar→Rw | Cl→Ar | Cl→Pr | Cl→Rw | Pr→Ar | Pr→Cl | Pr→Rw | Rw→Ar | Rw→Cl | Rw→Pr |    Avg   |
>     |:-------:|:-----:|:----:|:------:|:------:|:------:|:------:|:------:|:------:|:------:|:------:|:------:|:------:|:--------:|
>     |   NRC   |  57.4 | 79.6 |  81.5  |  67.8  |  80.2  |  78.9  |  65.1  |  56.7  |   83   |  71.3  |  58.3  |  84.7  | **72.1** |
>     | NRC+ELR |  58.4 |  79  |  81.7  |  70.5  |  79.7  |  79.5  |  66.7  |  58.2  |  82.8  |  73.4  |  60.3  |  85.3  |  **73**  |
>
>     In addition, we would like to emphasize that this neighborhood number-increasing trick exactly validates the neighborhood error accumulation problem that we mentioned in the Introduction. Specifically, in the early stage of the adaptation training, due to the existence of the unbounded label noise, there is an error accumulation problem in neighbors' label information. Therefore, NRC set the number of neighbors providing information to be smaller (3) in the first half of the training process. After some training steps, when the model's accuracy is guaranteed, NRC makes more use of the neighbors' label for the pseudo-label correction to boost the model's performance further.

---

> ### Author Response · Authors · 2022-11-14
> **Response to Reviewer 3S6g (2/4)**
>
> > **Q2**: **The proposed early learning regularization (ELR) term encourages the model prediction to stick to the early-time predictions for each data point. Meanwhile, NRC contains a loss term, called self-regularization loss, which has a similar purpose of reducing the potential impact of noisy pseudo-labels and not ignoring the current prediction. Please provide comparative explanations between the two and how the ELR still improves the NRC(Table1).**
>
> Thanks for your remark. We will explain the differences between ELR and the Self-Regularization (SR) term in NRC as follows:
>
>  - **Firstly, the motivation is different.** Even though both ELR and SR were originally designed to reduce noise impact, SR is **designed for neighbors' noisy mitigation** by **emphasizing the "ego feature" of current prediction**[1]. However, ELR aims to encourage the model prediction to stick to the **early-time predictions** and is designed for the label noise over the whole target data.
>  - **Besides, the prediction information involved in these two loss terms differs.** Given that $\mathcal{L}_\text{ELR}(\theta_t)=\log(1-\bar {y}_t^\top f(\mathbf{x};\theta_t))$ and $\mathcal{L}_\text{SR}(\theta_t)=-\hat{y}_t^\top f(\mathbf{x};\theta_t)$ with $\bar {y}_t=\beta \bar {y}_\text{t-1} + (1-\beta)f(\mathbf{x};\theta_t)$ and $\hat{y_t} = f(\mathbf{x};\theta_t)$, we can observe that **ELR involves the previous training step's prediction information** in loss (included in $\bar{y}_t$); however, **SR leverages only the prediction result of the current step**.
>  - **Finally, the effects of ELR and SR terms are also different.** By analyzing the gradient formulas of ELR and SR, we can find that for ELR, the gradient $\frac{d \mathcal{L}_\text{ELR}(\theta_t)}{d f(\mathbf{x};\theta_t)}$ increases as the model prediction closes to the target $\bar{y}_t$. And this will further force the prediction $f(\mathbf{x};\theta_t)$ close to $\bar{y}$ thanks to the large magnitude of the gradient, which will help with the utilization of early-time predictions and ETP. However, the gradient of SR, $\mathcal{L}_\text{SR}$, is a constant vector with values of prediction logits, which could be very small. When $\frac{d  \mathcal{L}_\text{SR}(\theta_t)}{d f(\mathbf{x};\theta_t)}$ is small, the SR term can be easily overwhelmed by the other loss terms that favor fitting incorrect pseudo labels, leading to poor performance. A similar analysis of how gradient affects the model performance in the presence of noisy labels can be found in Appendix C of [4].
>
> Therefore, the self-regularization term in NRC **does not exploit the ETP** and is fundamentally different from the ELR term we proposed. As a result, it can not handle the unbounded label noise in SFDA. We think this is why the performance of NRC+ELR can be improved. **We added a new section (Appendix I) with detailed explanations about this point in the manuscript**. We hope that these explanations will help the readers to have a clearer understanding of our novelty and contributions in the SFDA field.
>
>
> > **Q3**: **In Empirical Observations on Real-World Dataset, it seems the authors trained classifiers using the fixed noisy labels with cross-entropy loss. However, SFDA algorithms (e.g. NRC) gradually update pseudo-labels rather than using the fixed pseudo-labels. Please provide clarifications. It would be also nice if you can also show the training accuracy of SFDA algorithms in Figure 2.**
>
> Thanks for your insightful remark and advice.
>
> In Figure 2, we want to show the existence of ETP in the source-free domain adaptation setup while utilizing the classical cross-entropy loss. Therefore, we utilized the fixed labels without adopting any pseudo-label updating strategy, and the experimental results well verify our theoretical conclusion. They show that the CE loss and some conventional LLN losses like GCE suffer from the noise label memorizing problem in SFDA scenarios. We want to emphasize that Figure 2 is only used for the illustration purpose. In the experiments, we followed the standard SFDA pipelines and implemented the baseline methods with their proposed pseudo-label updating strategies.
>
> Meanwhile, we agree with your viewpoint that the current SFDA algorithms gradually update pseudo-labels, and they do not belong to the situations we described in Figure 2. This question is very insightful and interesting. So we re-conducted the experiments of NRC and NRC + ELR on Office-Home and drew similar "training step-accuracy" plots, **shown in Figure 12 in the newly added Appendix I.2**. We observe that thanks to the update of the pseudo-label during adaptation in the SFDA setting, overall, NRC can obtain a model with relatively high accuracy on the target domain. However, the performance drop still exists when using the NRC method alone, which can be effectively avoided by adding the ELR term. This confirms that ELR can effectively leverage ETP and avoid the problem of noisy label memorization.

---

> ### Author Response · Authors · 2022-11-14
> **Response to Reviewer 3S6g (1/4)**
>
> We would like to sincerely appreciate the reviewer 3S6g for the insightful review, detailed questions, and constructive suggestions. We will fully address the concerns below.
>
> > **Q1**: **While the previous SFDA methods do not explicitly formulate the SFDA problem as LLN, I think they are still partially based on the LLN methods. As the authors explained in the related work section, the SFDA methods leverage the target “noisy” pseudo labels and also some type of pseudo-label purification processes as in the LLN methods. It would be great if the authors can provide more clear explanations of the similarities and differences between their motivations.**
>
> Thanks for your comments. We would conclude the similarities and differences between the current SFDA and LLN methods as follows.
>
> 1. **The main similarity** between the existing SFDA approaches and the LLN methods is that both research fields have to **deal with data with noise**, **aiming to get a model with promising performance**.
>
> 2. As for **the differences**, they can be mainly divided into the following aspects.
>
>     - **From the perspective of motivations**, most of the existing SFDA approaches are developed under the domain adaptation setting. They study how to best exploit the distribution relationship between the source and target domains in the absence of source data, so as to achieve domain adaptation better. **Their motivation is to investigate how to better assign the pseudo-label.** In contrast, LLN is an independent field that mainly studies, given a set of noisy data, **how to deal with the label noise, conduct the model training, and obtain a noise-robust model with better performance.** Traditionally, the study of LLN does not involve assumptions about the data domain or source model. Meanwhile, there are more in-depth and rigorous studies (theoretical and methodological) on the types of noises, and how to handle and exploit them.
>     - **From the perspective of the methodology**, in order to obtain a higher quality pseudo-label, many SFDA methods **heuristically** use clustering or neighbor features to correct the pseudo-labels, and use the corrected labels to perform a normal supervised learning. The current SFDA methods focus on the **explicit pseudo-label purification process, which can be summarized as noisy label correction**. **However, for LLN, the noisy label correction is just a research sub-branch.** LLN also includes many other research directions, such as studies of different label noise types, research about how to utilize and even benefit from label noise in the training process, and how to train the model more robustly. Many noise-robust loss functions and related theoretical analyses have been developed. In this work, we focused on these more general LLN approaches and explored some new research methods for SFDA from them.
>
> We would like to emphasize that the motivation of our paper is to investigate how to study SFDA from the perspective of learning with label noise. We combine the characteristics of SFDA with the LLN approaches and discover the unbounded nature of label noise in SFDA. Further, we rigorously distinguish which LLN methods can help SFDA problems and which approaches are limited in their use in SFDA. We believe that the studies of LLN can open new avenues for the research of SFDA and bring more ideas and inspiration to the design of the SFDA algorithm.
>
> Thanks again for your valuable remark. **We have added the above discussions in Appendix J of the updated manuscript**.

---

### Official Review · Reviewer_CRh4 · 2022-10-25

**Confidence:** 3
**Correctness:** 3
**Technical Novelty And Significance:** 3
**Empirical Novelty And Significance:** 2
**Recommendation:** 6

**Clarity, Quality, Novelty And Reproducibility:**

This paper is novel to me. Although the designed method may not that practical, the theoretical analysis is important and can help design new algorithms for source-free domain adaptation.

**Details Of Ethics Concerns:**

I have not found any ethics concerns.

**Strength And Weaknesses:**

Strength:
+ The theoretical analysis that the label noise is unbound when there is a domain shift is novel to me.
+ The authors have demonstrated theoretically the early-time training phenomenon under certain conditions. Although the assumption can be strong, it is still interesting to see such an analysis.
+ This paper is well-motivated. It is important to understand the difference between two different but similar problem settings. It can be helpful for the designing of new algorithms for source-free domain adaptation.

Weakness:
+ The assumption for Theorem 4.1 that at most half of the samples are mislabelled could be a little bit strong. The proposed method based on the early-time training phenomenon may not be that practical.


**Summary Of The Paper:**

The authors have proved that the “label noise” in source-free domain adaptation is unbounded. Therefore, existing LLN methods that rely on their distribution assumptions are unable to address the “label noise” in source-free domain adaptation (SFDA). The authors have also demonstrated theoretically that the early-time training phenomenon can also be observed in the SFDA problem.

**Summary Of The Review:**

The major contribution of this paper is the theoretical analysis. The practical novelty may not be that significant. I would like to weakly accept this paper.

---

> ### Author Response · Authors · 2022-11-14
> **Response to Reviewer CRh4**
>
> We would like to sincerely appreciate the reviewer CRh4 for the detailed and insightful feedback! We hope the following responses can clarify your confusion, and we are happy to provide additional explanations if needed.
>
> > **Q1**: **The assumption for Theorem 4.1 that at most half of the samples are mislabelled could be a little bit strong. The proposed method based on the early-time training phenomenon may not be that practical.**
>
> Thanks for the remark. **On the one hand, we would like to explain that the "at most half mislabelled samples" assumption is reasonable.** To further illustrate this point, we added a plot as Figure 8 in Appendix B. From the plot, we can observe that the mislabeling rate over the whole target domain is always less than 0.5, which is an underlying property of the modeling. This illustrates that the assumption of theorem 4.1 is consistent with the conclusion of theorem 3.1 and will not conflict with the existence of unbounded label noise. **More importantly, the experimental results on real-world datasets can show the practicability of benefiting from ETP by ELR in SFDA.** For example, **from Table 1 to Table 4, we can observe that the source-only model is reasonably accurate**. This shows that in the SFDA problems, thanks to the correlation between the source and target domain distributions and a well-trained source domain model, the classification results of the source model on the target domain are not random guesses, even at the beginning of the adaptation. Therefore, a certain accuracy can be guaranteed, and the assumption can be satisfied. In fact, the observed performance drop for CE and GCE losses shown in Figure 2 and the experimental results in Table 1-4 for ELR can all demonstrate the existence of the ETP phenomenon in SFDA and the practicability of our proposed approach.
>
> Once again, we are grateful for your acknowledgment of our theoretical contributions, as well as the agreement on the potential new approaches that could be developed for SFDA research in the future.

---

> ### Author Response · Authors · 2022-11-18
> **Discussion**
>
> Dear Reviewer CRh4,
>
> We thank you again for your constructive suggestions, insightful comments, and valuable time.
>
> As the discussion deadline is approaching, we would appreciate it if you could let us know whether our responses have addressed your questions and concerns. We are happy to provide any further information or clarifications if needed.
>
> Best,
>
> Paper3266 Authors

---

### Official Review · Reviewer_RMmn · 2022-11-04

**Confidence:** 4
**Correctness:** 3
**Technical Novelty And Significance:** 3
**Empirical Novelty And Significance:** 2
**Recommendation:** 6

**Clarity, Quality, Novelty And Reproducibility:**

The work studied the LLN problem in SFDA, and showed that LLN is unbounded label noise, which had not been discussed before in previous literature. It theoretically and empirically proved that the early-time training phenomenon exists in the unbounded label noise scenario. The work actually is an extension of  ELR method (by Liu 2020)  on unbounded label noise. Their proposed early learning regularization (ELR) loss can be utilized on not only SFDA but also any LLN with unbounded noise.

**Details Of Ethics Concerns:**

nil

**Strength And Weaknesses:**

Strength:

This work theoretically and empirically justifies that the early-time training phenomenon exists in the unbounded label noise scenario, and proposed an extended ELR scheme for SFDA methods. The work has enough novelty for a practical problem. The paper provides reasonable experimental results and fair discussions. And the paper basically is well-written and easy to follow.

Weakness:

The paper lacks explanations and details in some places for readers who are not familiar with the topic. The paper should be more self-contained. According to the experiment results, the improvement by the proposed technique is marginal 1~2%.

In EQ1, should $d_1$ and $d_2$ be $d_1 = \frac{\mu_2 - \mu_1}{2} - C$ and $d_2 = \frac{\mu_2 - \mu_1}{2} + C$  ?
Please add a brief explanation of $l_{LLN} $ in Lemma 3.2 in the paper.






**Summary Of The Paper:**

Source-free domain adaptation (SFDA) methods usually use the pseudo-labels generated by source models which can be very noisy due to domain shifts. This paper studied and proposed an early learning regularization (ELR) based (Liu 2020) SFDA method to tackle the problem of learning with label noise (LLN). They show that pseudo-label noise generated by source models in SFDA is unbounded label noise. They theoretically and empirically proved that early-time training phenomenon exists in the unbounded label noise. Extensive expreiments were conducted to show that the proposed ELR based method can improve existing SFDA algorithms by around 1~2% on multiple DA benchmark datasets.

**Summary Of The Review:**

In summary, the work theoretically and empirically studied the LLN in SFDA and the $\textit{early-time training phenomenon}$ in unbounded LLN which hadn't been studied before. It proposed a ELR based solution for the unbounded LLN.  The work has enough novelty for the practical problem of training with unbounded label noise. The paper is basically well-written with fair experimental results.

---

> ### Author Response · Authors · 2022-11-14
> **Response to Reviewer RMmn (2/2)**
>
> >**Q4: Please add a brief explanation of $\ell_{LLN}$ in Lemma 3.2 in the paper.**
>
> Thanks for your remark and advice.
>
> - We would like to explain that the $\ell_{LLN}$ in Lemma 3.2 represents a set of noise-robust loss functions widely utilized in bounded LLN problems, which is an important branch of LLN research but has been proven not suitable for the unbounded noise in SFDA.
>
> - We want to clarify that in Appendix D of the original manuscript, we have given a brief description of the noise-robust loss based LLN methods and some concrete instances of $\ell_{LLN}$ involved in Lemma 3.2 before proving it. However, we agree that more details will allow people to understand our paper faster and better. Therefore, we **clarified the meaning of** $\ell_{LLN}$ **while introducing Lemma 3.2 in the main paper**. Besides, we **added a section in Appendix (D.1)** to give the specific forms of the noise-robust losses and to further present the related LLN methods in detail.
>
> **References:**
>
> [1] *Shiqi Yang, Yaxing Wang, Joost van de Weijer, Luis Herranz, Shangling Jui: Exploiting the Intrinsic Neighborhood Structure for Source-free Domain Adaptation. In NeurIPS 2021*
>
> [2] *Shiqi Yang, Yaxing Wang, Joost van de Weijer, Luis Herranz, Shangling Jui: Generalized Source-free Domain Adaptation. In ICCV 2021*
>
> [3] *Haifeng Xia, Handong Zhao, Zhengming Ding: Adaptive Adversarial Network for Source-free Domain Adaptation. In ICCV 2021*

---

> ### Author Response · Authors · 2022-11-14
> **Response to Reviewer RMmn (1/2)**
>
> We would like to sincerely appreciate the reviewer RMmn for the insightful review and constructive suggestions. We will fully address the concerns below.
>
> > **Q1**: **The paper lacks explanations and details in some places for readers who are not familiar with the topic. The paper should be more self-contained.**
>
> Thanks for your remark. We agree that more explanation and details would allow people unfamiliar with LLN or SFDA to understand our paper faster and better. Therefore, we have made the following adjustments to the paper (the revised and the added contents have been highlighted in blue):
>
> - To make the paper more self-contained, we adjusted the wording in the **Introduction** and the **Related Work** to better introduce the definition and background concepts of the related LLN methods. For example, we included the full names of the existing LLN methods mentioned in the Introduction section. Besides, we reformulated the related work of LLN by further categorizing and summarizing the existing LLN methods.
> - To better introduce the background of LLN and the LLN methods based on noise-robust losses, we **added a section in Appendix (D.1)**. Specifically, we presented some basic concepts and the concrete forms of the noise-robust losses involved in our paper.
>
> We hope this new version can clarify the unclear points and address your concerns. If you have further questions about this point, we kindly hope that you could provide more details. We would like to answer any questions and address your concerns further.
>
> > **Q2**: **According to the experiment results, the improvement by the proposed technique is marginal 1~2%.**
>
> Thanks for your comments. We would like to explain that the experimental results show that the performance improvement of ELR is actually **consistent and stable**.
>
> - On the one hand, to the best of our knowledge, a consistent improvement by 1~3% across 4 DA benchmarks and 3 different baseline methods is not a trivial result in SFDA research[1-3]. For example, the average performance improvement of NRC[1] compared with SHOT on Office-Home, Office-31 and VisDA is 1.4%; as for G-SFDA[2], the average improvement on Office-Home and VisDA is 1.25%. Therefore, we think our experimental results can support the utility of the proposed ELR technique.
> - Besides, we would like to emphasize that few SFDA methods have conducted experiments on DomainNet, a relatively new and challenging benchmark for DA. On top of the existing SFDA methods, ELR can still provide a performance improvement of 1.8-3%, which could indeed confirm the existence of the ETP phenomenon in SFDA and the effectiveness of ELR.
>
> > **Q3**: **In EQ1, should $d_1$ and $d_2$ be $d_1 = \frac{\mu_{2} - \mu_{1}}{2} - C$ and $d_2 = \frac{\mu_{2} - \mu_{1}}{2} + C$?**
>
> Thanks for the question. We think there is some misunderstanding regarding the definitions of $d_1$ and $d_2$ in EQ1.
>
> In the paper, we suppose $\mathcal{X}$ is a high dimensional space and therefore $\boldsymbol{\mu_{1}}, \boldsymbol{\mu_{2}}, \boldsymbol{c}$ are all **vectors** in $\mathbb{R}^{d}$, where $\boldsymbol{\mu_{1}}, \boldsymbol{\mu_{2}}$ are the $d$-dimensional mean vectors for the two-component Multivariate Gaussian Distribution, and $\boldsymbol{c}$ represents the projection of the domain shift $\mathbf{\Delta}$ on the vector $\boldsymbol{\mu_2}-\boldsymbol{\mu_1}$. However, $d_1$ and $d_2$ are two **scalar values** introduced in the process of calculating $ \Pr_{(\mathbf{x},y)\sim \mathcal{D}_T}[f_S(\mathbf{x})\not=y]$ as $d_1=||\frac{\boldsymbol{\mu_2}-\boldsymbol{\mu_1}}{2}-\mathbf{c}||\mathrm{sign}(||\frac{\boldsymbol{\mu_2}-\boldsymbol{\mu_1}}{2}||-||\mathbf{c}||)$ and $d_2=||\frac{\boldsymbol{\mu_2}-\boldsymbol{\mu_1}}{2}+\mathbf{c}||\mathrm{sign}(||\frac{\boldsymbol{\mu_2}-\boldsymbol{\mu_1}}{2}||+||\mathbf{c}||)$.
>
> Actually, we claim that **the absolute values of $d_1$ and $d_2$ are equal to the norms of $\frac{\boldsymbol{\mu_{2}} - \boldsymbol{\mu_{1}}}{2} - \boldsymbol{c}$ and $\frac{\boldsymbol{\mu_{2}} - \boldsymbol{\mu_{1}}}{2} + \boldsymbol{c}$**, respectively. Considering $d_2$ is always a positive scalar by calculation, we omit its sign expression and obtain the complete expressions for $d_1$ and $d_2$. More details of the calculation process for $d_1$ and $d_2$ are provided in Sec B.1 in Appendix B.
>
> To avoid potential ambiguity, we have changed the letter $C$, appearing in the following paragraph of Theorem 3.1, to $m$. We would like to claim that this letter represents an arbitrary constant with the meaning of the clean samples rate of each component, and there is no relation between it and $\boldsymbol{c}$. Besides, **an additional plot illustration of EQ1** has been added in Appendix B (shown in **Figure 8**). We hope it can help understand the EQ1 and the relationship between the mislabeling rate and the domain shift $\Delta$.

---

> ### Author Response · Authors · 2022-11-18
> **Discussion**
>
> Dear Reviewer RMmn,
>
> We thank you again for your constructive suggestions, insightful comments, and valuable time.
>
> According to your suggestions, we have included more details about the background concepts of LLN and the differences between the current LLN research and the SFDA methods in the revised manuscript. We hope these explanations can make the paper clearer and further clarify the novelties and contributions of our work.
>
> As the discussion deadline is approaching, we would appreciate it if you could let us know whether our responses have addressed your questions and concerns. We are happy to provide any further information or clarifications if needed.
>
>
> Best,
>
> Paper3266 Authors

---

### Decision · Program_Chairs · 2023-01-20

**Decision:**

Accept: notable-top-25%

**Justification For Why Not Higher Score:**

While of interest to the community, the submission does not have the kind of reviewer support to justify an oral presentation.

**Justification For Why Not Lower Score:**

While reviewers unanimously assigned a score of 6 to the submission, I don't see any major unaddressed reviewer concern, and the paper makes enough new contributions to be worth highlighting in a spotlight.

**Metareview: Summary, Strengths And Weaknesses:**

The submission looks into source-free domain adaptation (SFDA) from the perspective of learning with label noise (LLN). It shows that the label noise distribution in source-free domain adaptation is different from that of the usual LLN setting, making LLN approaches unsuitable for SFDA. However, it demonstrates that the early-time training phenomenon (ETP) observed in LLN is also present in SFDA, and that using ETP to address label noise in SFDA yields significant improvements to existing SFDA approaches on four benchmarks (Office-31, Office-Home, VisDA, and DomainNet).

Reviewers find the submission clear (RMmn, 3S6g, nSg6), and the questions they raised in that regard were appropriately addressed by the authors in the rebuttal (Reviewer 3S6g on the connection between SFDA and LNN and the description of the baselines and experimental protocol; Reviewer nSg6 on theoretical and empirical results). Reviewers also note the submission's convincing theoretical and empirical contributions. Opinions were originally split on novelty: Reviewers RMmn found the paper novel enough and the theoretical analysis was new to Reviewer CRh4; Reviewer 3S6g was concerned that although leveraging ETP for SFDA is new, the mechanism for doing so is borrowed from existing work (ELR) and shares similarities with NRC. The authors' explanations addressed Reviewer 3S6g's concerns.

All reviewers now agree that the paper meets the bar for acceptance; I therefore recommend acceptance.

**Note From Pc:**

if the above contains the word "oral" or "spotlight" please see: "oral" presentation means -> notable-top-5% and "spotlight" means -> notable-top-25%. As stated in our emails, we are disassociating presentation type from AC recommendations

**Summary Of Ac-Reviewer Meeting:**

N/A